# OCCUQUEST: MITIGATING OCCUPATIONAL BIAS FOR INCLUSIVE LARGE LANGUAGE MODELS

## ABSTRACT

The emergence of large language models (LLMs) has revolutionized natural language processing tasks. However, existing instruction-tuning datasets suffer from occupational bias: the majority of data relates to only a few occupations, which hampers the instruction-tuned LLMs to generate helpful responses to professional queries from practitioners in specific fields. To mitigate this issue and promote occupation-inclusive LLMs, we create an instruction-tuning dataset named *OccuQuest*, which contains 110,000+ prompt-completion pairs and 30,000+ dialogues covering over 1,000 occupations in 26 occupational categories. We systematically request ChatGPT, organizing queries hierarchically based on Occupation, Responsibility, Topic, and Question, to ensure a comprehensive coverage of occupational specialty inquiries. By comparing with three commonly used datasets (Dolly, ShareGPT, and WizardLM), we observe that OccuQuest exhibits a more balanced distribution across occupations. Furthermore, we assemble three test sets for comprehensive evaluation, an occu-test set covering 25 occupational categories, an estate set focusing on real estate, and an occu-quora set containing real-world questions from Quora. We then fine-tune LLaMA on OccuQuest to obtain OccuLLaMA, which significantly outperforms state-of-the-art LLaMA variants (Vicuna, Tulu, and WizardLM) on professional questions in GPT-4 and human evaluations. Notably, on the occu-quora set, OccuLLaMA reaches a high win rate of 86.4% against WizardLM. Furthermore, we demonstrate the potential of combining OccuQuest with other instruction-tuning datasets to enhance the overall performance of LLMs. By fine-tuning LLaMA on a mixture of OccuQuest and Tulu datasets, we introduce ProLLaMA, which excels in addressing occupational questions and exhibits superior performance in comprehensive evaluations such as MMLU, GSM8K, BBH, and HumanEval. Among the different LLaMA variants, the 7B and 13B ProLLaMA models achieve the highest performance on MMLU and GSM8K, with the 7B ProLLaMA model demonstrating an improvement of more than 4 points over the other 7B variants on GSM8K.

## 1 INTRODUCTION

The emergence of large language models (LLMs), such as GPT (Brown et al., 2020; Ouyang et al., 2022; OpenAI, 2023), PaLM (Chowdhery et al., 2022; Chung et al., 2022), LLaMA (Touvron et al., 2023) and its various variants, trigger a paradigm shift in natural language processing (NLP) tasks. Instruction tuning has become a crucial process following pre-training, aiming to align the behavior of LLMs with human expectations. However, we observe a notable **occupational bias** in the existing instruction-tuning datasets, as a significant portion of data is centered around specific occupational groups. Unfortunately, language models tend to capture and reflect this bias (Suresh & Guttag, 2021; Shen et al., 2022; Lee et al., 2023), making it challenging for them to generate accurate and insightful responses to questions from specific occupations.

The primary origins of instruction-tuning data encompass pre-existing NLP tasks (Flan (Wei et al., 2022), SuperNI (Wang et al., 2022b)), manually constructed instructions (Dolly (Conover et al., 2023), OpenAssistant (Köpf et al., 2023)), and datasets generated using LLMs (Alpaca (Taori et al., 2023), ShareGPT[1], WizardLM (Xu et al., 2023a)). Practitioners in the AI industries are more likely

---

[1] https://sharegpt.com/

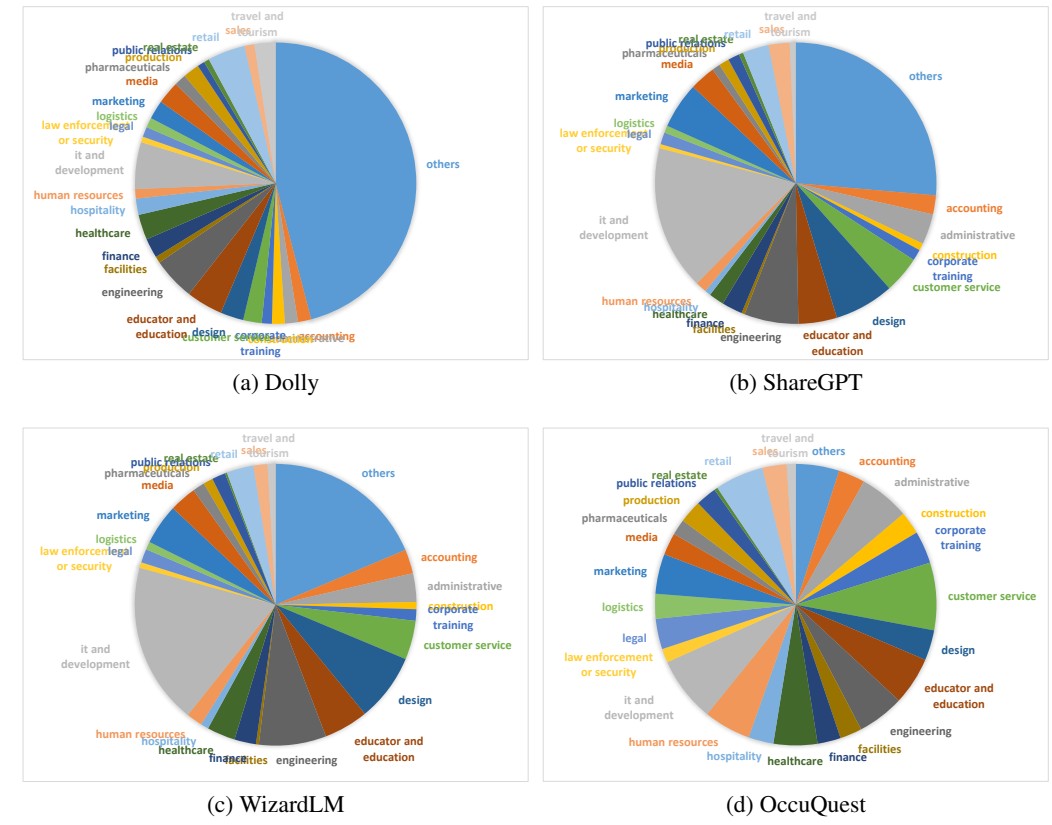

Figure 1: The distribution of occupational categories across various datasets.

to access these sources, as Databricks admits that the Dolly dataset comes from over 5,000 employees who are very interested in LLMs (Conover et al., 2023). However, practitioners across various fields with weak connections to AI communities have limited access to these data sources. A welder stands less chance of producing instruction-tuning data than an employee of an AI institute. Consequently, while the LLMs fine-tuned on these datasets excel in answering queries related to building chatbots, they may struggle with questions about rectifying the lack of fusion in welding. These allocative harms (Barocas et al., 2017) hamper the LLMs from providing helpful, honest, and harmless (Askell et al., 2021) assistance to specific occupational groups.

To create more inclusive and unbiased language models that can better serve users from different occupational backgrounds, we propose an instruction-tuning dataset that covers over 1,000 occupations. We collect over 1,000 job titles and their responsibilities spanning 26 distinct occupation categories from Workable[2]. We illustrate the categories and the representative occupations in Appendix A. We then utilize ChatGPT[3] to identify key topics of concern for practitioners in each occupation and generate relevant questions and answers accordingly. This effort results in the creation of a comprehensive dataset called OccuQuest comprising 148,772 queries and responses, covering 1,013 occupations and 31,811 topics.

We compare the distribution of occupational categories in OccuQuest with three typical instruction-tuning datasets (Dolly, ShareGPT, and WizardLM) using ChatGPT, and Figure 1 illustrates the results. The precise distribution percentages are presented in Appendix F. Our analysis reveals that Dolly, ShareGPT, and WizardLM favor non-occupation-related topics (denoted as *Others*) and the "IT and Development" category, while OccuQuest exhibits a more balanced distribution. For instance, ShareGPT and WizardLM consist of less than 0.8% of data in the "Facilities" category, while

---

[2]https://resources.workable.com/job-descriptions/
[3]In the study, we use the gpt-3.5-turbo API (https://platform.openai.com/docs/models/gpt-3-5) because it is capable and fee friendly.

comprising over 15% of data in "IT and Development". Conversely, in OccuQuest, the majority of occupational categories encompass data ranging from 2% to 6%.

To validate the effectiveness of OccuQuest, we fine-tune LLaMA on OccuQuest to get OccuLLaMA and compare it with the state-of-the-art LLaMA variants (Vicuna (Chiang et al., 2023), WizardLM, and Tulu (Wang et al., 2023a)) through preference assessments using GPT-4[4] and human evaluations. OccuLLaMA consistently outperforms other variants in answering occupational questions across various occupations. Notably, on a test set consisting of real-world questions covering 25 occupational categories, OccuLLaMA achieves an 86.4% win rate against WizardLM.

Moreover, we demonstrate that the OccuQuest dataset can be effectively combined with other instruction-tuning datasets to enhance the comprehensive abilities of LLMs. Following Wang et al. (2023a), we fine-tune LLaMA on a mixture of OccuQuest and Tulu datasets to obtain ProLLaMA. ProLLaMA excels in addressing occupational questions and performs well in comprehensive ability evaluations such as MMLU (Hendrycks et al., 2021), GSM8K (Cobbe et al., 2021), BBH (Suzgun et al., 2023), and HumanEval (Chen et al., 2021). When compared to the above LLaMA variants, the 7B and 13B ProLLaMA models achieve the best performance on MMLU and GSM8K. In particular, on GSM8K, the 7B ProLLaMA surpasses these 7B variants by a margin exceeding 4 points.

In summary, this article makes four main contributions:

1. We propose the OccuQuest dataset, which consists of 148,772 queries and responses covering 1,013 occupations. To the best of our knowledge, this is the first dataset available that focuses on mitigating the issue of occupational bias in LLMs.

2. We demonstrate the effectiveness of OccuQuest through preference tests with GPT-4 and human evaluations. Additionally, we showcase the integration of OccuQuest with existing datasets to enhance LLMs in a synthetic manner.

3. We propose ProLLaMA, a series of LLaMA models that excel in answering questions from different occupations and perform well on the comprehensive abilities assessments.

4. We openly release our dataset and model parameters, encouraging further research and exploration in this domain.

## 2 RELATED WORKS

### 2.1 BIAS IN DATASETS

The utilization of deep neural networks relies heavily on datasets, yet existing datasets contain a wide variety of biases including race (Manzini et al., 2019; Sambasivan et al., 2021; Lee et al., 2023; Field et al., 2023), gender (Koolen & van Cranenburgh, 2017; Rudinger et al., 2018), disability (Hutchinson et al., 2020; Gadiraju et al., 2023), and others. Extensive researches uncover and analyze these biases in traditional NLP tasks (Vanmassenhove et al., 2019; Henderson et al., 2018). Additionally, there is a growing recognition of the social implications and consequences of these biases (Hovy & Spruit, 2016; Barocas et al., 2017).

One prominent and effective approach to address biases involves constructing or transforming the datasets. For instance, Costa-jussà & de Jorge (2020) and Saunders & Byrne (2020) fine-tune models on carefully screened and balanced data to mitigate biases. Wang et al. (2022a) adopt data augmentation strategies by randomly switching entities to prevent the translation system from associating specific names with contextual idiosyncrasies. Choubey et al. (2021) generate gender-specific pseudo-parallel corpora to prompt translation systems to produce accurate gender-specific translations. Motivated by the insights from these studies, we aim to mitigate occupational bias in LLMs by constructing an occupationally balanced instruction-tuning dataset. To the best of our knowledge, this is the first endeavor specifically targeting the mitigation of occupational bias in LLMs.

### 2.2 INSTRUCTION TUNING

In recent years, there has been significant progress in LLMs, with notable advancements demonstrated by GPT-3 (Brown et al., 2020), highlighting the potential of context learning in LLMs. As

---

[4]https://platform.openai.com/docs/models/gpt-4

a result, numerous LLMs have emerged, such as Gopher (Rae et al., 2021), Chinchilla (Hoffmann et al., 2022), and PaLM (Chowdhery et al., 2022), showcasing exceptional performance and dominance across diverse NLP tasks.

To align the behavior of LLMs with human preferences, instruction tuning has emerged as a crucial method (Ouyang et al., 2022; Chung et al., 2022). There are three primary sources of existing instruction-tuning datasets: datasets derived from pre-existing NLP tasks, datasets created through manual authoring, and datasets generated using LLMs. Initially, instruction-tuning datasets are developed by expanding upon existing NLP task datasets. For example, Flan (Wei et al., 2022) and SuperNI (Wang et al., 2022b) are designed by converting data from diverse NLP tasks, such as classification, extraction, and infilling, into instructions, inputs, and outputs format using templates. However, using existing datasets has the drawback of limited diversity in topics and syntax within the instructions. To overcome this limitation, Dolly (Conover et al., 2023) and OpenAssistant (Köpf et al., 2023) enhance diversity by manually crafting prompts. Additionally, recent studies have explored cost-effective and efficient approaches by leveraging ChatGPT to obtain prompts and responses, reducing the costs and labor involved (Wang et al., 2023b; Taori et al., 2023; Xu et al., 2023a; Chiang et al., 2023; Xu et al., 2023b).

Extensive efforts have been dedicated to augmenting instruction-tuning datasets, aiming to improve the generalization capabilities of LLMs. However, these existing datasets exhibit a limited occupational distribution, resulting in inadequate precision and granularity of responses to queries from specific occupations. This study aims to construct an instruction-tuning dataset that encompasses a wide range of occupation-related topics, thereby mitigating the occupational bias present in LLMs.

## 3  OCCUQUEST DATASET

### 3.1  DATASET CONSTRUCTION

To mitigate the issue of occupational bias in the instruction-tuning corpus, we intend to construct a dataset that encompasses a wide range of occupational specializations. We request ChatGPT hierarchically, focusing on Occupation, Responsibility, Topic, and Questions, to cover as many occupations and their corresponding areas of interest as possible. The data construction process consists of five steps, which are outlined below.

**Step 1: get occupations.** To begin, we gather occupation titles and their associated responsibilities. Workable offers more than 1,000 occupational titles organized into 26 occupational categories. Each occupation is accompanied by a list of responsibilities, consisting of one sentence per responsibility. We successfully collect 1,037 occupations and their respective responsibilities from Workable, with an average of around 7 responsibilities per occupation.

**Step 2: request topics.** We utilize ChatGPT to generate multiple related topics and topic features by providing the occupation name and one responsibility. A topic is a keyword or keywords that reflect what a practitioner needs to consider when fulfilling a specific responsibility and topic features provide a descriptive paragraph about the topic. To avoid duplication, we employ MinHash (Broder, 2000) on topic features to filter out topics that exhibit high similarities.

**Step 3: request prompts.** Using the topic and topic features obtained in Step 2, we request ChatGPT to generate multiple prompts describing potential queries that practitioners may encounter. During the request, ChatGPT is asked to list the keywords and then generate the prompts to produce diverse prompts with distinct keywords, as directly generating prompts tends to result in similar prompts. Same as the topic filtering process, we filter out the prompts that show high similarities.

**Step 4: get responses.** In this step, we ask ChatGPT to answer the prompts generated in Step 3. To improve the accuracy of the completions, we assign ChatGPT a role corresponding to the occupation before some of the queries. We also remove responses that contain overly similar completions.

**Step 5: create dialogs.** After completing Step 4, we have data for a single round of queries and responses in the OccuQuest dataset. To enhance the model's ability to handle multi-round requests, we additionally request ChatGPT to generate multi-round dialogues between a rookie and a veteran, discussing problem-solving scenarios encountered at work for each topic.

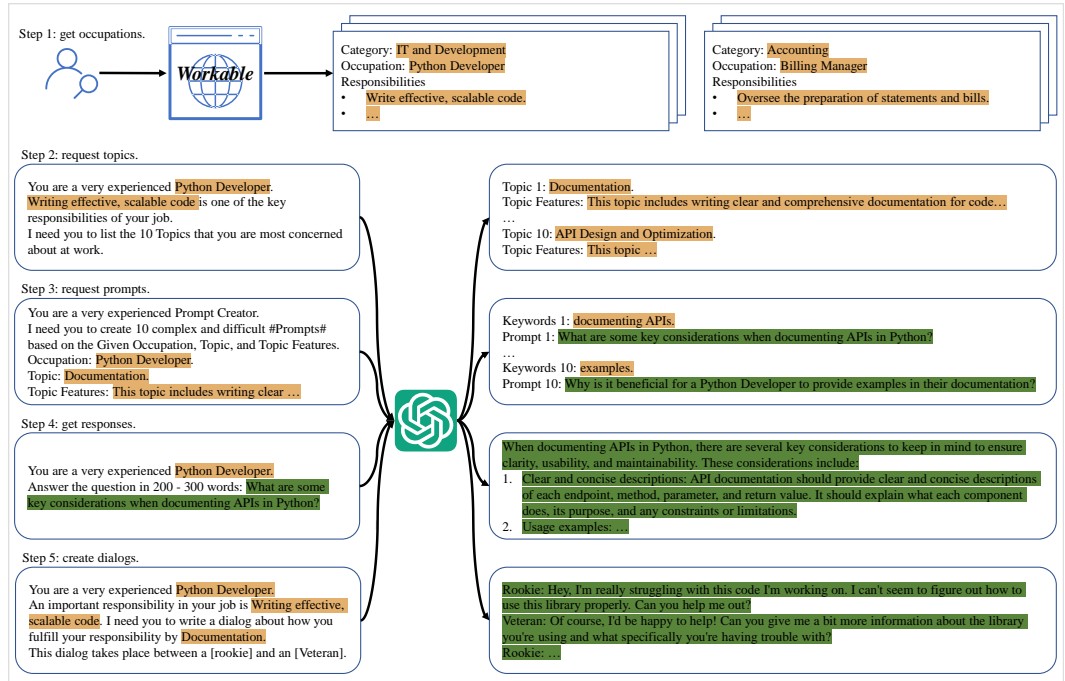

Figure 2: An illustration of the OccuQuest dataset construction process, where the contents highlighted with a background color are ultimately gathered to constitute the dataset. To eliminate duplicate samples, MinHash filtering is applied after steps 2, 3, and 4.

During Steps 2, 3, 4, and 5, we exclude responses that are less than 50 words in length or contain the phrase "Sorry, as an AI assistant..." to ensure the validity of the responses. We incur an approximate cost of $300 for API access fees in the dataset construction process.

Figure 2 illustrates an example of the dataset construction process, while the actual prompts used to collect the dataset can be found in Appendix E. The examples extracted from OccuQuest are provided in Appendix B.

## 3.2 DATASET SPLIT

To assess the efficacy of OccuQuest and the models, we partition a portion of the data from OccuQuest as test sets. Specifically, we designate the data within the "Real estate" category as the holdout set. From this category, we randomly select 250 samples, referred to as the "estate set", to evaluate the models' generalization capabilities. In the remaining 25 categories, we randomly select 100 samples and 10 samples from each as the validation set and "occu-test" set, respectively. The remaining data in these 25 categories are allocated for the training set. To ensure the evaluation aligns closely with real-world scenarios, we collect 250 authentic questions (10 questions per category, totaling 25 categories) from Quora[5] as the "occu-quora" set.

In summary, OccuQuest consists of the following components:

1. A training set, containing 114,090 prompt-completion pairs and 31,682 dialogues across 25 categories;
2. A validation set, containing 2,500 prompt-completion pairs across 25 categories;
3. An occu-test set, containing 250 prompt-completion pairs across 25 categories;
4. An estate set, containing 250 prompt-completion pairs in the "Real estate" category;
5. An occu-quora set, containing 250 real-world questions gathered from Quora across 25 categories.

---

[5]https://www.quora.com/

### 3.3 BALANCED DISTRIBUTION OF OCCUPATIONS

OccuQuest contains 117,090 prompt-completion pairs and 31,682 multi-round dialogues, encompassing 1,013 occupations under 26 occupational categories. Each item in OccuQuest contains the occupational category, occupation name, topic, topic features, queries, and responses. Compared to the existing instruction-tuning datasets, OccuQuest exhibits a balanced distribution of occupations.

To evaluate the distribution of occupations within OccuQuest compared to existing instruction-tuning datasets, we select three prominent datasets for comparison: Dolly, ShareGPT, and WizardLM. Dolly is manually authored, ShareGPT is obtained by the users interacting with ChatGPT, and WizardLM is generated by expanding existing instructions using ChatGPT. These datasets represent the primary sources of current instruction-tuning data. We randomly select 10,000 samples from each of the datasets and inquire ChatGPT about the occupational category to which each sample is likely to belong. The specific prompt used for this task can be found in Appendix E.

The results are presented in Figure 1. In Dolly, ShareGPT, and WizardLM, the "Others" category unrelated to specific occupations dominates the distribution. Furthermore, the categories of "IT and Development" and "Engineering" also exhibit a disproportionately high proportion compared to other occupations, consistent with our claim in Section 1 that individuals from these fields are more likely to contribute data. In contrast, OccuQuest demonstrates a more balanced distribution of occupational categories, without any single category displaying clear dominance. For detailed percentages of occupation distribution across different datasets, please refer to Appendix F.

## 4 EXPERIMENTS

### 4.1 BASELINES

We fine-tune the LLaMA-7B model on OccuQuest and compare it to competitive baselines:

**Vicuna**, an open-source chatbot trained by fine-tuning LLaMA on ShareGPT. Preliminary GPT-4 evaluation reveals that Vicuna-13B achieves over 90% quality compared to OpenAI ChatGPT (Chiang et al., 2023). We utilize the checkpoint available on the Huggingface model repository[6].

**Tulu**, a fine-tuned LLaMA on a combination of existing instruction-tuning datasets, including FLAN, Dolly, OpenAssistant, Alpaca, and ShareGPT, proposed by Wang et al. (2023a). Tulu achieves the best average performance on several benchmarks including MMLU, GSM8K, BBH, etc. We utilize the checkpoint available on the Huggingface model repository[7].

**WizardLM**, a LLaMA fine-tuned with complex instructions derived from extending the seed instructions in the Alpaca dataset using ChatGPT. WizardLM achieves more than 90% capacity of ChatGPT on 17 out of 29 evaluated skills (Xu et al., 2023a). We utilize the checkpoint available on the Huggingface model repository[8].

**ChatGPT**, a chatbot proposed by OpenAI, recognized as one of the most powerful LLMs (services) currently available. We use the gpt-3.5-turbo API[9] provided by OpenAI for our experiments.

### 4.2 TRAINING DETAILS

To obtain the OccuLLaMA model, we fine-tune the LLaMA-7B model using the OccuQuest training set. The fine-tuning process involves training for 5 epochs, with a batch size of 128 and a total of 5,500 training steps. We employ the AdamW optimizer with a maximum learning rate of $2 \times 10^{-5}$, and the learning rate is linearly decayed during training. Additionally, we set the warmup ratio to 0.03 to gradually increase the learning rate at the beginning of training. The entire training process is executed on a server equipped with $8 \times 80G$ A100 GPUs and completes within 8 hours.

---

[6]`https://huggingface.co/lmsys/vicuna-7b-v1.3`
[7]`https://huggingface.co/allenai/tulu-7b`
[8]`https://huggingface.co/WizardLM/WizardLM-7B-V1.0`
[9]`https://platform.openai.com/docs/models/gpt-3-5`

### 4.3 EVALUATION SETUP

We generate the responses to the queries in the occu-test, estate, and occu-quora sets employing the baselines and OccuLLaMA through greedy search, with the maximum generation length set to 1024 tokens. Subsequently, we evaluate the responses using GPT-4 and human evaluations.

#### 4.3.1 GPT-4 EVALUATION

The evaluation of open-ended generation using LLMs highlights the benefits of scalability and explainability, and previous studies have shown that GPT-4 exhibits high agreement with human experts (Zheng et al., 2023). Therefore, we leverage GPT-4[10] to evaluate the performance of OccuLLaMA and the baselines in addressing occupation-related queries. During this evaluation, we compare the responses from OccuLLaMA with those generated by each baseline. To ensure fairness and avoid any positional bias, we judge each query twice by swapping the order of the two responses and only declare a win when a response is preferred in both orderings (Zheng et al., 2023). For the specific prompt utilized in the evaluation, please refer to Appendix E.

#### 4.3.2 HUMAN EVALUATION

We conduct a human evaluation to assess the alignment of the generated responses with human expectations. Due to the substantial labor costs associated with human evaluation, we randomly select two questions from each occupational category in the occu-test and occu-quora sets, resulting in a human evaluation set comprising 100 samples. We engage three annotators who are tasked with rating the responses on three dimensions: **Helpfulness**, **Honesty**, and **Harmlessness** (Askell et al., 2021). These dimensions are assessed on a scale of 1 to 5, with higher scores indicating superior performance. For more details about the human evaluation, please refer to Appendix G.

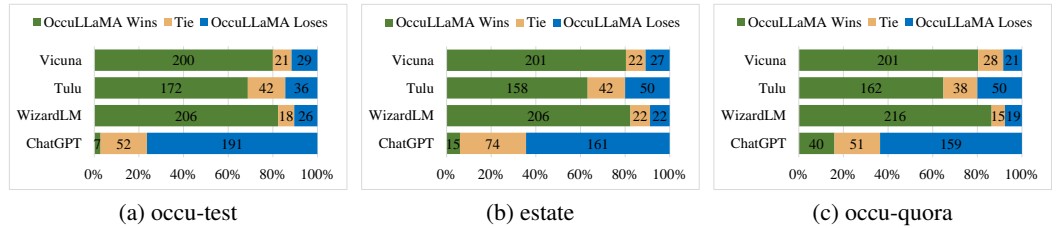

|            | (a) occu-test | (b) estate | (c) occu-quora |
|------------|---------------|------------|----------------|

Figure 3: GPT-4 evaluation results on OccuLLaMA against the comparative baselines.

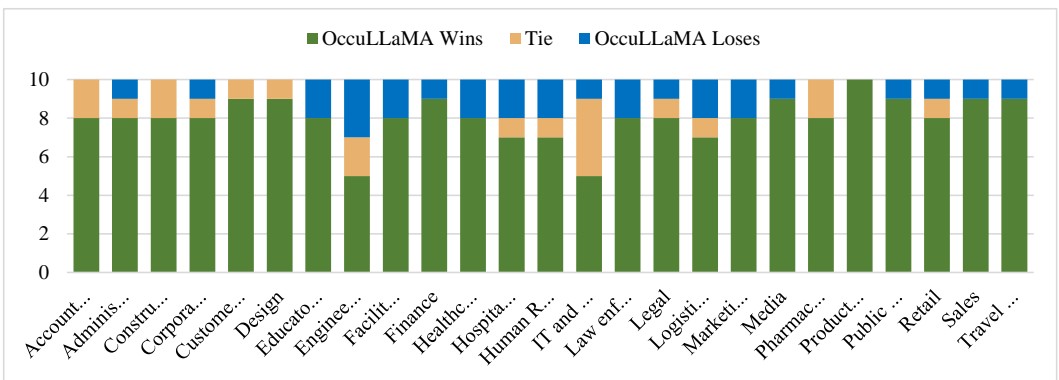

Figure 4: The win rates of OccuLLaMA vs Vicuna under different occupational categories.

---

[10]We use the "gpt-4" API in `https://platform.openai.com/docs/models/gpt-4`.

## 4.4 EXPERIMENTAL RESULTS

Figure 3 illustrates the results of GPT-4 evaluation. The findings clearly indicate that **OccuLLaMA outperforms other LLaMA-based models in answering occupation-related questions across all three evaluation sets**. In comparison to Vicuna and WizardLM, OccuLLaMA consistently achieves high win rates, exceeding 80%. When compared to Tulu, OccuLLaMA consistently achieves win rates of over 60% and failure rates of under 20% across the test sets. These results highlight the superiority of OccuLLaMA in effectively addressing occupation-related questions, underscoring the effectiveness of OccuQuest in enhancing the occupational capabilities of LLMs.

However, it is evident that there still exists a significant disparity between OccuLLaMA and ChatGPT. We propose that this discrepancy may be attributed to two factors: a) The limited capabilities of the base LLaMA model, making it challenging to comprehensively enhance its performance with a limited amount of instruction-tuning data; b) The OccuQuest dataset is derived from ChatGPT.

Figure 4 presents the win rates of OccuLLaMA against Vicuna across various occupational categories in the occu-test set. **OccuLLaMA demonstrates a significant advantage over Vicuna in all occupational categories.** Notably, OccuLLaMA exhibits relatively weaker strength in the fields of "Engineering" and "IT and Development", which aligns with the observed distribution of occupations in the ShareGPT dataset, wherein a substantial portion of the data pertains to the "Engineering" and "IT and Development" domains. Similar patterns can be observed in Tulu and WizardLM, and we provide the win rates of OccuLLaMA against these models in Appendix C.

Table 1 presents the results of the human evaluation. Notably, **in terms of "Helpfulness", OccuLLaMA demonstrates performance comparable to ChatGPT and significantly outperforms other LLaMA variants**. Regarding "Honesty" and "Harmlessness," except for Vicuna, which performs poorly, the evaluated models exhibit similar performance. This observation may be attributed to the absence of harmful or misleading questions in the test set. The results of the human evaluation further affirm the superiority of OccuLLaMA in accurately addressing occupation-related questions, highlighting the efficiency of the OccuQuest dataset in mitigating the occupational bias of LLMs.

We provide examples of the generated responses in Appendix D.

Table 1: Human evaluation results. We use Fleiss' Kappa (Fleiss, 1971) to measure the inter-rater agreement and the agreement scores falling within 0.40-0.60 indicate "moderate agreement".

|            | **Helpfullness** | **Honesty** | **Harmlessness** |
| ---------- | ---------------- | ----------- | ---------------- |
| Vicuna     | 3.79             | 4.15        | 4.65             |
| Tulu       | 4.05             | 4.75        | 4.82             |
| WizardLM   | 4.19             | 4.73        | 4.86             |
| ChatGPT    | 4.57             | 4.83        | 4.90             |
| OccuLLaMA  | 4.45             | 4.77        | 4.88             |
| Agreement  | 0.48             | 0.42        | 0.55             |

## 4.5 COMBINING WITH OTHER DATASETS

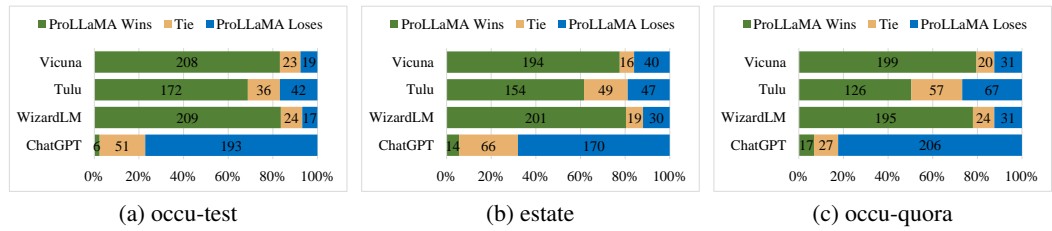

(a) occu-test     (b) estate     (c) occu-quora

Figure 5: GPT-4 evaluation results on ProLLaMA against the comparative baselines.

Table 2: Evaluation results on comprehensive benchmarks, with the top-performing results highlighted in **bold**.

| Model | MMLU (0 shot) | GSM8K (8 shot) | BBH (cot) | HumanEval (P@10) |
|---|---|---|---|---|
| Vinalla LLaMA-7B | 30.8 | 9.9 | 32.5 | 16.2 |
| Vicuna-7B | 44.4 | 16.1 | 34.6 | 14.8 |
| Tulu-7B | 44.4 | 26.8 | **37.0** | 20.6 |
| WizardLM-7B | 36.1 | 14.9 | 31.8 | 20.7 |
| ProLLaMA-7B | **46.2** | **31.2** | 35.5 | **21.2** |

The OccuQuest dataset is specifically designed to address occupational queries, but it is limited in coverage of reasoning abilities, such as mathematical skills. Following Wang et al. (2023a), we fine-tune LLaMA on a mixture of the Tulu dataset and OccuQuest to get ProLLaMA. The training process of ProLLaMA is similar to OccuLLaMA and takes approximately 50 hours.

We evaluate ProLLaMA's occupational proficiency on OccuQuest and assess its comprehensive abilities using established benchmarks, including MMLU (Hendrycks et al., 2021) for world knowledge, GSM8K (Cobbe et al., 2021) for mathematical reasoning ability, BBH (Suzgun et al., 2023) for general reasoning capabilities, and HumanEval (Chen et al., 2021) for coding skills.

Figure 5 provides the preference evaluation results obtained using GPT-4. **ProLLaMA exhibits similar performance to OccuLLaMA in answering occupational questions, surpassing the other LLaMA variants by a significant margin.** Table 2 shows the results on benchmarks. **ProLLaMA outperforms the other variants significantly on MMLU and GSM8K, with improvements of over 1.8 and 4.4 points on MMLU and GSM8K respectively, while demonstrating comparable performance on BBH and HumanEval.** A plausible explanation for the enhanced MMLU results is the inclusion of specific fields in MMLU relating to occupations, for instance, the field "health" is closely associated with "Healthcare". The improved performance on GSM8K can potentially be attributed to the mathematical data present in OccuQuest, where fields like "Accounting" and "Marketing" are prominent. Moreover, the incremental problem-solving approach adopted in various occupations contributes to the enhancement of LLM's reasoning abilities.

We provide the experimental results of the 13B models in Appendix H, where similar superiority can be observed. These findings highlight the effectiveness of OccuQuest in mitigating occupational bias without sacrificing the reasoning enhancements provided by other datasets.

## 5 CONCLUSION

The current data available for instruction-tuning is plagued by occupational bias that a significant portion of the data is only relevant to a few professions. Consequently, this limitation hinders the ability of models trained on such data to effectively handle queries from individuals with specific professional backgrounds. To mitigate this issue and develop more inclusive and unbiased large language models, we create the OccuQuest dataset. This dataset encompasses a wide range of topics associated with over 1,000 occupations. A comparison with existing instruction-tuning datasets like Dolly (Conover et al., 2023), ShareGPT, and WizardLM (Xu et al., 2023a) reveals that OccuQuest exhibits a much more balanced distribution across different occupations. We fine-tune LLaMA on OccuQuest to obtain OccuLLaMA. Through GPT-4 and human evaluations, OccuLLaMA demonstrates superiority over state-of-the-art LLaMA variants in effectively answering professional queries related to various occupations. OccuQuest can also be effectively combined with other instruction-tuning datasets to enhance the overall capabilities of large language models. By fine-tuning LLaMA on both OccuQuest and Tulu datasets, we develop ProLLaMA, which excels in answering occupational questions and proves advantageous in comprehensive ability evaluations, including MMLU, GSM8K, BBH, and HumanEval. To summarize, our contributions consist of the creation of the OccuQuest dataset, the validation of its efficacy through preference tests using GPT-4 and human evaluations, the proposal of OccuLLaMA and ProLLaMA models, and the open release of our dataset and model parameters for further research. These endeavors are aimed at fostering more inclusive and unbiased language models that can better cater to users from diverse occupational backgrounds. Furthermore, we discuss the limitations of this study in Appendix I.

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

## A    OCCUPATIONAL CATEGORIES

Table 3: The occupational categories and the corresponding representative occupations within each respective category.

| Category | Typical Occupations | Category | Typical Occupations |
|---|---|---|---|
| Accounting | Cost analyst, Tax Preparer | IT and Development | UX Researcher, Computer Science |
| Administrative | Non-Profit Executive Director, Physicist | Law enforcement or Security | Parole Officer, Deputy Sheriff |
| Construction | Solution architect, Lineman | Legal | Notary, Duty Clerk |
| Corporate training | Stockbroker, Technical Training Manager | Logistics | Program Analyst, Driver |
| Customer service | Customer Education Specialist, Mail Carrier | Marketing | Copy Editor, Channel Partner Manager |
| Design | Physical Product Designer, Product Designer | Media | Movie Makeup Artist, Film Director |
| Educator and Education | Registrar, Child Care Provider | Pharmaceuticals | Chemist, Clinical Pharmacist |
| Engineering | Meter Reader, Product Engineer | Production | Product Analyst, Car Detailer |
| Facilities | Air Traffic Controller, Groundskeeper | Public Relations (PR) | Grant Writer, Public Relations Assistant |
| Finance | VP of Finance, Portfolio Manager | Real estate | Real Estate Agent, Leasing Agent |
| Healthcare | Nurse Manager, Chief Medical Officer | Retail | Beauty Advisor, Cashier |
| Hospitality | Coroner, Valet | Sales | Target Cashier, Sales Clerk |
| Human Resources (HR) | Comp Analyst, Employee Relations | Travel and Tourism | Arborist, Cabin Crew |

Table 3 shows the occupational categories we collect and some of the representative occupations under each category. More details about each category and the related occupations can be found in the Workable website[11].

---

[11] https://resources.workable.com/job-descriptions/

## B  DATA EXAMPLES IN OCCUQUEST

Figure 7 to Figure 12 present several examples extracted from the OccuQuest dataset. The items within OccuQuest are available in two distinct formats: prompt-completion pairs and dialogues. The prompt-completion pairs consist of various components, including occupational category, occupation, topic, topic features, prompt, and completion. The completion segment accurately and comprehensively addresses the question presented in the prompt. The dialogues encompass occupational category, occupation, topic, topic features, and multiple rounds of conversations between a rookie and a veteran. Throughout the conversation, the veteran guides the rookie step-by-step in refining the problem and ultimately aids in its resolution, demonstrating commendable initiative and guidance.

## C  WIN RATES UNDER DIFFERENT OCCUPATIONAL CATEGORIES

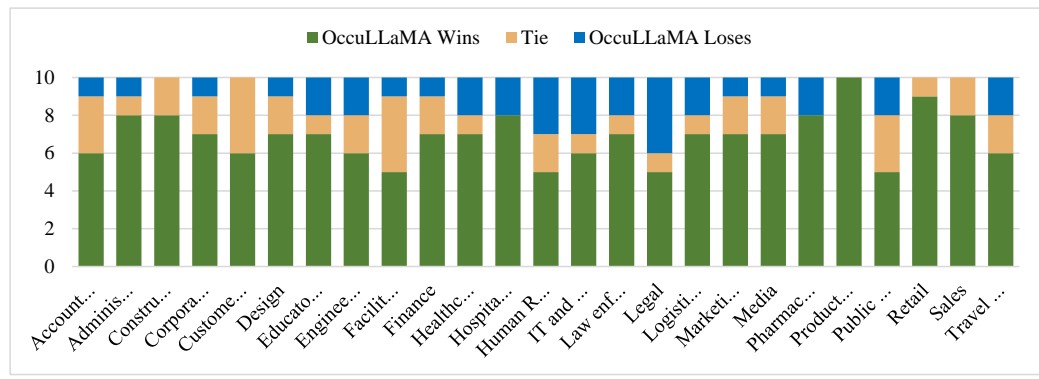

(a) OccuLLaMA vs Tulu

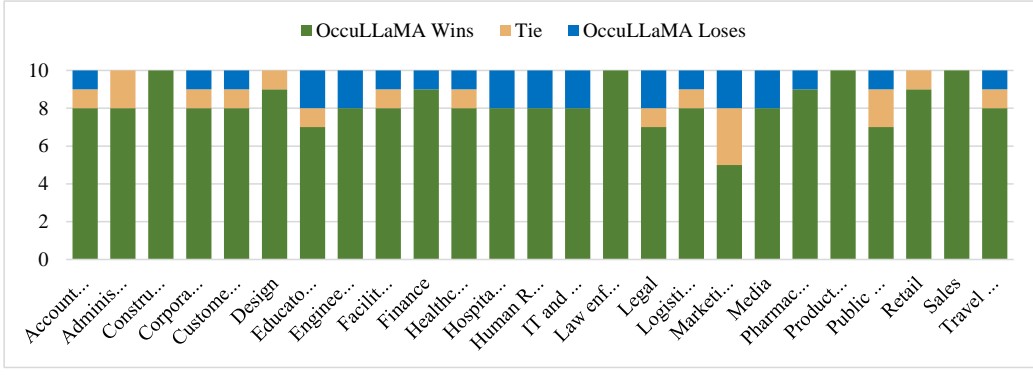

(b) OccuLLaMA vs WizardLM

Figure 6: The win rates of OccuLLaMA vs Tulu and WizardLM under different occupational categories.

Figure 6 illustrates the win rates of OccuLLaMA compared to Tulu and WizardLM in the GPT-4 evaluation across various occupational categories. As discussed in Section 4.4, our analysis reveals that OccuLLaMA demonstrates significant advantages across all occupational categories, with notable expertise in categories such as Construction and Production, which suffer from limited data availability in the prevailing instruction-tuning datasets.

## D  GENERATED RESPONSE EXAMPLES

Figures 13 through Figure 16 depict the responses generated by different models in relation to occupation-related questions. OccuLLaMA and ChatGPT tend to produce more detailed and ac-

curate responses when compared to Vicuna, Tulu, and WizardLM. Moreover, OccuLLaMA usually provides specific examples, such as "gaps, cracks, or incomplete penetration" and "rust, oil, or dirt" in Figure 13 and "medical history, medication information, or personal identification details" in Figure 14, to enhance user comprehension. Notably, the responses from Vicuna often contain errors such as abrupt switches to unrelated topics (Figure 13) and continuation of conversations (Figure 16).

Comparatively, OccuLLaMA's responses encompass essentially the entirety of the content, similar to ChatGPT, with some variations in the level of detail and emphasis. For instance, in Figure 15, both ChatGPT and OccuLLaMA provide comprehensive and accurate answers. However, ChatGPT focuses on the "process of researching historical fashion trends," whereas OccuLLaMA emphasizes "how it influences the work of a Costume Designer". This may explain why OccuLLaMA scores very close to ChatGPT in the human evaluation.

Additionally, Figure 17 and Figure 18 showcase two examples of multi-turn conversation with OccuLLaMA. These examples illustrate OccuLLaMA's ability to provide accurate responses to both professional and daily questions, as well as its demonstration of good initiative.

## E    WHOLE PROMPTS

Figure 19 to Figure 26 present the template prompts employed in this study. In practice, the words in {Braces} in the templates are replaced by corresponding information. To introduce a variety of complexity and form within the questions, we use three different prompts. In practice, for each topic, we randomly choose one from the prompts in Figure 20, Figure 21, and Figure 22. Lastly, the prompt employed for GPT-4 evaluation is sourced from the work of Zheng et al. (2023).

## F    DATASET STATISTICS

Table 5 presents the occupational distribution of various datasets. In Dolly, a significant portion of the data (45.9%) is labeled as "Others" and is not related to specific occupations. "IT and Development" and "Engineering" each constitute around 5% of the data, while categories such as "Real estate," "Law enforcement or Security," "Facilities," and "Public Relations" each account for less than 1% of the data. ShareGPT and WizardLM exhibit similar distributional patterns to Dolly. In contrast, OccuQuest displays a more balanced distribution, with most categories representing approximately 2 to 5% of the data. However, it is worth noting that the "Real estate" category still has relatively low amounts of data. Consequently, we utilize the data in the "Real estate" category as a holdout test set to assess the generalization capability of our models. The results presented in Section 4.4 demonstrate that our models outperform other LLaMA variants in addressing "Real estate" queries.

Table 6 illustrates the number of occupations, topics, and data within each occupational category in OccuQuest. There are substantial variations in the number of occupations across different categories. For instance, the Healthcare category encompasses 119 occupations, while Pharmaceuticals and Real estate only consist of 6 occupations. These discrepancies contribute to the slightly uneven distribution of occupations in OccuQuest. Additionally, we provide the average number of topics and data associated with each occupation, which demonstrates a relatively equal distribution.

Table 7 presents the statistics for OccuQuest and several instruction-tuning datasets. OccuQuest comprises two distinct components: prompt-completion pairs and dialogues. The prompt-completion pairs consist of concise instructions and detailed responses, which are intended to assist LLMs in efficiently addressing specialized user queries. On the other hand, the dialogues involve multi-round conversations, with a significantly higher average number of rounds compared to other datasets. We employ this dialogue data to enable the LLMs to guide users to better comprehend and resolve challenges through multiple interactions. The LLMs fine-tuned using OccuQuest exhibit the ability to comprehensively answer user questions and effectively handle multi-round conversations, actively guiding users through step-by-step interactions to solve problems. Several exemplary generated responses are presented in Appendix D.

## G  HUMAN EVALUATION DETAILS

We use human evaluation to assess whether the models meet human expectations when answering occupation-related questions. The models we compare include OccuLLaMA and four baselines, Vicuna, Tulu, WizardLM, and ChatGPT. We randomly select 2 questions from each occupational category in the occu-test set and the occu-quora set to form a human evaluation set containing 100 samples. Three annotators are asked to rate the responses in terms of **Helpfulness**, **Honesty**, and **Harmlessness** (Askell et al., 2021) on a scale of 1-5, the higher the better. We present detailed scoring guidelines and inform all the annotators of the potential risks caused by the negative statements generated by artificial intelligence. The annotators are recruited in the authors' labs and they all have relevant English publications in the field of neural networks. Payment for this human evaluation is made in full by the lab's supervisor based on the workload. Figure 27 illustrates the guideline page for human evaluation, and Figure 28 shows an example in the evaluation.

## H  EVALUATION RESULTS ON 13B MODELS

Table 4: Evaluation results of 13B models on comprehensive benchmarks, with the top-performing results in 13B models highlighted in **bold**. ChatGPT and GPT-4 are closed-source models with undisclosed number of parameters, and their results are listed here for reference.

| Model | MMLU (0 shot) | GSM8K (8 shot) | BBH (cot) | HumanEval (P@10) |
|---|---|---|---|---|
| Vinalla LLaMA-13B | 39.3 | 16.5 | 37.0 | 21.1 |
| Vicuna-13B | **49.2** | 26.2 | 40.8 | 24.9 |
| Tulu-13B | **49.2** | 35.4 | **43.2** | 23.8 |
| WizardLM-13B | 48.9 | 34.0 | 36.6 | **31.7** |
| ProLLaMA-13B | **49.2** | **36.3** | 39.7 | 23.1 |
| ChatGPT | 67.9 | 76.0 | 66.1 | 88.4 |
| GPT-4 | 82.4 | 92.5 | 88.0 | 94.1 |

Table 4 shows the experimental results of the 13B models on the comprehensive benchmarks. Similar to the experimental results of the 7B models, ProLLaMA demonstrates the best results on MMLU and GSM8K. WizardLM shows amazing results on HumanEval, far surpassing other models, which may be related to the fact that the dataset it uses contains complex code-related instructions. ProLLaMA utilizes data close to Tulu, and on these benchmarks, ProLLaMA shows comparable performance to Tulu, which is consistent with our conclusion that OccuQuest mitigates occupational bias without sacrificing the reasoning enhancements provided by other datasets.

## I  LIMITATIONS

OccuQuest is aimed at mitigating the issue of occupational bias in large language models by providing more occupationally equitable data. However, there are certain limitations that still exist in this research, according to the current knowledge.

In terms of the dataset's occupational balance, although OccuQuest surpasses other datasets in this aspect, there is still a significant lack of data in certain occupational categories, such as Real estate.

On data accuracy, it is important to note that there may be errors present. This is because ChatGPT is utilized for data collection, and it is inevitable that factual errors and hallucinations may occur in the ChatGPT system.

In terms of evaluation, the human evaluation only involves feedback from three annotators. Consequently, it does not encompass all occupations included in the test sets. This limitation may deviate from the preferences of actual professional practitioners.

Regarding language, the experiments are exclusively conducted in English. This may introduce a language bias in the large language models. However, the process proposed for constructing the OccuQuest dataset is not dependent on language, and it is believed that this process can be applied to other languages as well.

## J  ETHICS STATEMENT

We declare that all of the authors of this paper are aware of and honor the ICLR Code of Ethics. In this paper, we propose a method for constructing occupationally balanced instruction-tuning data by querying a large language model and will release our dataset and model parameters. In terms of methodology, our data construction method does not bring any ethical issues. We are open to other researchers to utilize and expand upon our method.

Regarding the dataset, there are three points that warrant attention: a) The collection of occupation titles and occupational responsibilities data is obtained from Workable[12], a source that holds copyright over this data. Consequently, any commercial utilization of this data necessitates prior authorization from Workable. b) The topics, prompts, responses, and dialogues are collected through OpenAI APIs[13], and this data is prohibited from being used to develop commercial products that compete with OpenAI. c) The topics, prompts, responses, and dialogues are generated using artificial intelligence, which implies that the data may contain offensive and biased elements. Regrettably, at present, we lack an effective means of comprehensively reviewing all the data. We strongly advise researchers to exercise caution when utilizing the dataset and models.

We conduct a human evaluation to rate the generated responses, and in doing so, we ensure that no personal information of any of the annotators is involved in this work. Furthermore, prior to the human evaluation, we meticulously review the samples to ensure the absence of any negative statements. All annotators are duly informed in advance about the potential risks associated with offensive statements generated by artificial intelligence.

---

**Category:** Accounting     **Occupation:** Accountant     **Topic:** Access Control Policies

**Topic features:**

  This topic involves implementing access control policies to restrict access to financial data based on the principle of least privilege. It includes assigning user roles and permissions, regularly reviewing access rights, and revoking access for terminated employees. By enforcing access control policies, an Accountant can prevent unauthorized access to financial data and minimize the risk of internal data breaches.

**Prompt:**

  Analyze the potential risks and consequences of unauthorized access to financial data and how access control policies implemented by Accountants can mitigate these risks.

**Completion:**

  Unauthorized access to financial data can pose significant risks to an organization. It can lead to financial fraud, identity theft, reputational damage, and regulatory non-compliance. Access control policies implemented by Accountants can mitigate these risks by ensuring that only authorized individuals have access to sensitive financial data. These policies can include strong password requirements, two-factor authentication, regular access reviews, and segregation of duties. By implementing these controls, Accountants can limit the potential for unauthorized access, detect any suspicious activities, and prevent unauthorized individuals from manipulating or misusing financial data. This helps to safeguard the integrity, confidentiality, and availability of financial information, protecting the organization from potential financial losses and legal consequences.

---

Figure 7: An example of the prompt-completion pair on Accountant in OccuQuest.

---

[12] https://resources.workable.com/job-descriptions/
[13] https://platform.openai.com/

---

**Category:** Construction    **Occupation:** Architect    **Topic:** Building Codes and Regulations Compliance

**Topic features:**

This topic requires a thorough understanding of building codes, regulations, and permit requirements. It involves collaborating with construction professionals to ensure that the project complies with all applicable codes and regulations, obtaining necessary permits, and addressing any compliance issues that may arise.

**Prompt:**

What strategies do architects employ to address compliance issues that arise during construction?

**Completion:**

Architects employ several strategies to address compliance issues that arise during construction. Firstly, they ensure that all construction documents and plans are in accordance with local building codes and regulations. They also conduct regular site visits to monitor construction progress and ensure compliance with the approved plans. If any compliance issues arise, architects work closely with contractors and other stakeholders to find solutions that meet both regulatory requirements and the project's design intent. Additionally, architects may collaborate with specialized consultants or experts to address specific compliance issues, such as fire safety or accessibility. Overall, architects play a crucial role in navigating compliance issues and ensuring that construction projects meet all necessary regulations and standards.

---

Figure 8: An example of the prompt-completion pair on Architect in OccuQuest.

---

**Category:** IT and Development    **Occupation:** Python Developer    **Topic:** Documentation

**Topic features:**

This topic includes writing clear and comprehensive documentation for code, APIs, and libraries. It involves documenting code functionality, usage instructions, and examples. By focusing on documentation, a Python Developer can make their code more accessible to other developers and facilitate easier adoption and maintenance.

**Prompt:**

Why is it beneficial for a Python Developer to provide examples in their documentation?

**Completion:**

Firstly, examples help to clarify the usage and functionality of the code or library being documented. By providing real-world scenarios and practical demonstrations, developers can better understand how to implement the code in their own projects.

Secondly, examples serve as a learning tool for beginners or those unfamiliar with the codebase. They can help new developers grasp the concepts and syntax more easily, reducing the learning curve and enabling them to start using the code effectively.

Furthermore, examples can act as a troubleshooting resource. When encountering issues or bugs, developers can refer to the examples to identify potential problems or compare their own implementation with the documented one. This can save time and effort in debugging and resolving issues.

Additionally, examples foster collaboration and knowledge sharing within the developer community. By providing clear and well-documented examples, developers can contribute to the open-source community and encourage others to build upon their work.

In conclusion, examples in documentation greatly enhance the usability, understandability, and collaborative potential of Python code, making it easier for developers to utilize and contribute to the codebase.

---

Figure 9: An example of the prompt-completion pair on Python Developer in OccuQuest.

**Category:** Accounting    **Occupation:** Accountant    **Topic:** Access Control Policies

**Topic features:**

This topic involves implementing access control policies to restrict access to financial data based on the principle of least privilege. It includes assigning user roles and permissions, regularly reviewing access rights, and revoking access for terminated employees. By enforcing access control policies, an Accountant can prevent unauthorized access to financial data and minimize the risk of internal data breaches.

**Dialog:**

*Rookie:* Hey, I'm glad I found you. I'm having a problem at work and I could really use your help.

*Veteran:* Of course, I'm here to assist you. What seems to be the issue?

*Rookie:* Well, I've been assigned the task of reinforcing financial data confidentiality and conducting database backups. I understand the importance of it, but I'm not sure how to implement access control policies effectively.

*Veteran:* Access control policies are indeed crucial in maintaining data confidentiality. Let's start with the principle of least privilege. It means that users should only have the minimum access necessary to perform their job functions. To implement this, you need to assign user roles and permissions based on their responsibilities.

*Rookie:* That makes sense. So, how do I go about assigning these roles and permissions?

*Veteran:* Firstly, identify the different job roles within your organization that require access to financial data. For example, you may have accountants, auditors, and managers. Then, determine the specific tasks and functions each role should be able to perform. Once you have this information, you can assign appropriate permissions to each role in the database management system.

*Rookie:* Okay, I think I understand. But how do I ensure that access rights are regularly reviewed and updated?

*Veteran:* Regularly reviewing access rights is crucial to maintain data security. You should establish a schedule to review user access rights, ideally on a quarterly or annual basis. This involves working closely with the IT department to generate a list of all users and their associated permissions. Then, you can compare this list against the current job roles and responsibilities to identify any discrepancies or unnecessary access.

*Rookie:* That sounds like a good plan. But what about employees who leave the company? How do we revoke their access?

*Veteran:* When an employee leaves the company, it's essential to promptly revoke their access to financial data. This can be done by working closely with the HR department to ensure that they notify you immediately when an employee is terminated. Once you receive this information, you should disable or delete their user account in the database management system to prevent any unauthorized access.

*Rookie:* That's really helpful. Is there anything else I should keep in mind while implementing access control policies?

*Veteran:* Absolutely. It's important to regularly monitor and log access attempts to financial data. This can be done using tools like intrusion detection systems or database activity monitoring software. Additionally, ensure that all employees are educated about the importance of data confidentiality and the consequences of unauthorized access.

*Rookie:* Thank you so much for your guidance. I feel much more confident now in implementing access control policies to reinforce financial data confidentiality and conduct database backups.

*Veteran:* You're welcome! Remember, access control is a continuous process, so always stay vigilant and adapt your policies as needed. If you have any more questions, don't hesitate to reach out. Good luck!

Figure 10: An example of the dialog on Accountant in OccuQuest.

**Category:** Construction      **Occupation:** Architect      **Topic:** Building Codes and Regulations Compliance

**Topic features:**

   This topic requires a thorough understanding of building codes, regulations, and permit requirements. It involves collaborating with construction professionals to ensure that the project complies with all applicable codes and regulations, obtaining necessary permits, and addressing any compliance issues that may arise.

**Dialog:**

*Rookie:* Hey, I'm glad I found you. I'm facing a problem at work and I could really use your expertise.

*Veteran:* Of course, I'm here to help. What seems to be the issue?

*Rookie:* Well, I'm working on a new project and I'm having trouble ensuring compliance with the building codes and regulations. I'm not quite sure where to start and how to navigate through all the requirements.

*Veteran:* Ah, building codes and regulations can be quite overwhelming, especially for someone new to the field. Don't worry, I'll guide you through it. First things first, have you thoroughly reviewed the local building codes and regulations applicable to your project?

*Rookie:* Yes, I have a copy of the codes, but I'm finding it difficult to understand all the technical jargon and how it applies to our design.

*Veteran:* That's completely normal. Building codes can be complex, but they are crucial for ensuring the safety and functionality of our structures. Let's start by breaking it down. Identify the specific sections of the codes that are relevant to your project, such as fire safety, structural requirements, accessibility, and so on.

*Rookie:* Okay, I'll go through the codes again and identify the relevant sections. Once I have that, what should I do next?

*Veteran:* Once you have identified the relevant sections, cross-reference them with your design plans. Make sure your design meets or exceeds the minimum requirements specified in the codes. If you come across any conflicts or uncertainties, consult with the appropriate construction professionals, such as structural engineers or fire safety experts.

*Rookie:* That makes sense. I'll reach out to the relevant professionals to ensure our design aligns with the codes. But what about permits? How do I know which ones are required for our project?

*Veteran:* Good question. Permit requirements vary depending on the location and the scope of the project. Check with the local building department or permit office to determine the specific permits needed. They will provide you with the necessary application forms and guide you through the process. Remember, it's crucial to obtain all the required permits before starting any construction work.

*Rookie:* Alright, I'll contact the local building department to get a clear understanding of the permits we need. What if we encounter compliance issues during construction? How should we handle them?

*Veteran:* Compliance issues can arise, but it's important to address them promptly. If you notice any deviations from the approved plans or if an inspector identifies non-compliance, document the issue and consult with the relevant construction professionals. Together, you can develop a plan to rectify the issue while ensuring compliance with the codes. Keep in mind that safety should always be the top priority.

*Rookie:* I see. So, it's crucial to maintain open communication with the construction professionals throughout the project to address any compliance issues that may arise.

*Veteran:* Absolutely. Collaboration and communication are key. Regularly meet with the construction team, attend site visits, and address any concerns promptly. Remember, building codes and regulations are in place to protect the public and ensure the quality of our work. By actively engaging with construction professionals and diligently following the codes, we can create safe and compliant structures.

*Rookie:* Thank you so much for your guidance. I feel much more confident now in navigating the complexities of building codes and regulations. I'll make sure to follow your advice and seek help whenever needed.

*Veteran:* You're welcome! Don't hesitate to reach out if you have any more questions or need further assistance. Building codes and regulations can be challenging, but with experience and guidance, you'll become more proficient in ensuring compliance. Good luck with your project!

Figure 11: An example of the dialog on Architect in OccuQuest.

**Category:** IT and Development    **Occupation:** Python Developer    **Topic:** Documentation

**Topic features:**

   This topic includes writing clear and comprehensive documentation for code, APIs, and libraries. It involves documenting code functionality, usage instructions, and examples. By focusing on documentation, a Python Developer can make their code more accessible to other developers and facilitate easier adoption and maintenance.

**Dialog:**

*Rookie:* Hey, I'm really struggling with this code I'm working on. I can't seem to figure out how to use this library properly. Can you help me out?

*Veteran:* Of course, I'd be happy to help! Can you give me a bit more information about the library you're using and what specifically you're having trouble with?

*Rookie:* I'm using a Python library called "PyData" for data manipulation, but I'm having a hard time understanding how to use some of the functions and classes it provides. The documentation is a bit confusing to me.

*Veteran:* Ah, I see. Well, one of the most important aspects of being a Python developer is writing clear and comprehensive documentation. It not only helps others understand your code but also makes it easier for you to maintain and update it in the future. So, let's start by looking at the documentation for the PyData library.

*Rookie:* Okay, but what should I be looking for in the documentation?

*Veteran:* When reading the documentation, pay attention to the code functionality, usage instructions, and examples. These will give you a good understanding of how to use the library effectively. Look for sections that explain the purpose of each function or class, the parameters they accept, and the expected return values. Additionally, examples can provide you with practical usage scenarios.

*Rookie:* That makes sense. But what if the documentation is not clear or lacks examples?

*Veteran:* If you find the documentation lacking, don't worry. There are a few things you can do. First, try searching for tutorials or blog posts online that demonstrate the usage of the library. Often, developers share their experiences and provide code examples that can be helpful. Additionally, you can explore the library's source code directly. By examining the code, you can gain insights into how the functions and classes are implemented.

*Rookie:* That's a great suggestion. But what if I still can't figure it out?

*Veteran:* If you're still struggling, don't hesitate to reach out to the library's community or the developer who maintains it. They are usually quite helpful and can provide you with guidance or even update the documentation based on your feedback. Remember, documentation is a collaborative effort, and your input can help improve it for others.

*Rookie:* I never thought about reaching out to the community or the developer. That's a good idea. I'll give it a try. Thanks for your help!

*Veteran:* You're welcome! Remember, writing effective and scalable code goes hand in hand with writing clear and comprehensive documentation. It not only helps you understand your own code but also makes it accessible to other developers. Good luck with your project, and don't hesitate to ask if you have any more questions!

Figure 12: An example of the dialog on Python Developer in OccuQuest.

**Category**: Construction     **Occupation**: Welder     **Topic**: Welding Troubleshooting and Problem-solving

**Query**: How can a Welder identify and rectify the issue of lack of fusion in welding?

Vicuna:

To identify and rectify the issue of lack of fusion in welding, a Welder can follow these steps:

1. Inspect the weld: The first step is to inspect the weld to identify the area where the lack of fusion is occurr The 1992 presidential campaign was one of the most contentious in American history. The incumbent, George H. W. Bush, was challenged by Bill Clinton, the governor of Arkansas. The campaign was marked by a number of controversies, including allegations of sexual impropriety by Clinton and questions about his character and fitness for office.

Bush, a Republican, had served as vice president under Ronald Reagan from 1981 to 1989. He was elected president in his own right in 1988, defeating Democratic candidate Michael Dukakis. Bush was a moderate Republican who had a reputation for being a strong leader on foreign policy issues. He was also known for his sense of humor and his ability to connect with ordinary Americans…

Tulu:

A welder can identify and rectify the issue of lack of fusion in welding by following these steps:

1. Check the welding parameters: The welder should check the welding parameters such as the welding current, voltage, and wire feed speed to ensure that they are set correctly for the material being welded.

2. Use the right welding technique: The welder should use the right welding technique, such as the correct welding angle and the correct amount of welding rod, to ensure that the weld is properly applied.

3. Monitor the weld: The welder should monitor the weld during the welding process to ensure that it is progressing as expected and that the weld is not overheating or undergoing excessive distortion.

4. Use the right filler material: The welder should use the right filler material, such as a welding rod or a welding wire, to ensure that the weld is properly fused.

5. Clean the weld: The welder should clean the weld to remove any contaminants that may have interfered with the fusion process.

6. Re-weld the area: If the welder is unable to achieve fusion in a particular area, they should re-weld the area and try again until the fusion is achieved.

By following these steps, a welder can identify and rectify the issue of lack of fusion in welding and ensure that the weld is properly fused.

WizardLM:

There are several ways a welder can identify and rectify the issue of lack of fusion in welding. Here are a few steps that can be taken:

1. Inspect the weld: The welder should inspect the weld to determine if there is any lack of fusion. This can be done by looking at the weld from different angles and checking for any gaps or voids.

2. Check the welding process: The welder should also check the welding process to ensure that it was done correctly. This includes checking the welding rod, the welding machine, and the welding tips.

3. Adjust the welding parameters: If the welder determines that the welding process was done correctly, but there is still a lack of fusion, they can adjust the welding parameters such as the welding speed, voltage, and amperage.

4. Use a different welding process: If the welder is still having trouble with lack of fusion, they can try using a different welding process such as MIG or TIG welding.

5. Use a different welding material: If the welder is still having trouble with lack of fusion, they can try using a different welding material such as a different type of steel or a different welding rod.

By following these steps, a welder can identify and rectify the issue of lack of fusion in welding.

ChatGPT:

To identify and rectify the issue of lack of fusion in welding, a welder can follow these steps:

1. Visual Inspection: The welder should visually inspect the weld joint to identify any signs of lack of fusion. Look for incomplete penetration, lack of fusion between the base metal and the weld metal, or lack of fusion between multiple weld passes.

2. Clean the Joint: Ensure that the joint is clean and free from any contaminants such as rust, oil, paint, or dirt. Use appropriate cleaning methods like wire brushing, grinding, or chemical cleaning to remove any surface impurities.

3. Adjust Welding Parameters: Check the welding parameters such as voltage, current, wire feed speed, and travel speed. Lack of fusion can occur if the heat input is too low or if the travel speed is too fast. Adjust the parameters to ensure proper heat input and penetration.

4. Preheat the Base Metal: In some cases, lack of fusion can be due to the base metal being too cold. Preheating the base metal before welding can help improve fusion. Use a preheating method suitable for the specific material being welded.

5. Increase Welding Technique: If lack of fusion persists, the welder can try different welding techniques. For example, using a weaving motion or backstepping technique can help ensure better fusion by increasing the heat input and allowing better mixing of the base metal and weld metal.

6. Use Proper Welding Consumables: Ensure that the welding consumables, such as electrodes or filler wires, are suitable for the base metal and welding process being used. Using the correct consumables can improve fusion and overall weld quality.

7. Post-Weld Inspection: After making the necessary adjustments, inspect the weld again to ensure that the lack of fusion issue has been rectified. Look for proper fusion between the base metal and weld metal, as well as between multiple weld passes.

If the lack of fusion issue persists despite these steps, it may be necessary to consult with a welding supervisor or expert for further guidance and troubleshooting.

OccuLLaMA:

As an experienced welder, I can identify and rectify the issue of lack of fusion in welding through several steps. Firstly, I would carefully inspect the weld joint to determine if there is any visible lack of fusion. This can be done by examining the weld bead for any gaps, cracks, or incomplete penetration.

If lack of fusion is identified, the next step is to assess the welding parameters. I would check the welding voltage, amperage, and travel speed to ensure they are within the appropriate range for the specific welding process and material being used. Improper settings can lead to insufficient heat input, resulting in lack of fusion.

Additionally, I would examine the welding electrode or filler material. Faulty or contaminated electrodes can cause lack of fusion. Cleaning the electrode and replacing it if necessary can often resolve the issue.

Another potential cause of lack of fusion is improper joint preparation. I would carefully inspect the joint for any contaminants, such as rust, oil, or dirt, and clean it thoroughly. Proper beveling and chamfering of the joint edges can also help ensure proper fusion.

In some cases, lack of fusion may be due to a welding technique mistake, such as incorrect angle or lack of proper travel speed. I would review my welding technique and make any necessary adjustments to ensure proper penetration and fusion.

If the above steps do not resolve the issue of lack of fusion, I would consult with a more experienced welder or supervisor for further guidance. They may suggest additional inspections, tests, or adjustments to rectify the problem.

In conclusion, identifying and rectifying the issue of lack of fusion in welding requires a thorough inspection of the weld joint, evaluation of welding parameters, examination of electrodes, proper joint preparation, and reviewing welding techniques.

Figure 13: The responses produced by distinct instruction-tuned models in response to an identical Welder related query.

**Category**: Healthcare    **Occupation**: Patient Service Representative    **Topic**: Accessibility of Forms

**Query**: In what situations would it be more appropriate to provide patients with paper forms rather than electronic ones?

Vicuna:

Paper forms may be more appropriate in situations where patients have limited access to technology, such as in rural or low-income areas, or for patients who prefer to fill out forms manually. Paper forms may also be more appropriate for forms that require a lot of writing or for forms that patients need to keep for their records. However, in most cases, electronic forms are more convenient and efficient for both patients and healthcare providers.

Tulu:

There are several situations where providing patients with paper forms may be more appropriate than electronic ones:

1. Limited internet access: If patients do not have access to the internet or do not have a device to access electronic forms, providing them with paper forms may be more convenient.

2. Technical difficulties: Electronic forms may be prone to technical difficulties, such as glitches or crashes, which can cause frustration and delays. Providing patients with paper forms can help ensure that they are able to complete the forms without any technical issues.

3. Legibility: Some patients may have difficulty reading or completing electronic forms due to visual impairments or other issues. Providing them with paper forms can help ensure that they are able to complete the forms accurately and easily.

4. Security concerns: Some patients may be concerned about the security of their personal information when completing electronic forms. Providing them with paper forms can help alleviate these concerns and provide a more secure option.

5. Limited literacy: Some patients may have limited literacy skills, which can make it difficult for them to complete electronic forms. Providing them with paper forms can help ensure that they are able to complete the forms without any difficulties.

Overall, providing patients with paper forms can be a more flexible and accessible option in certain situations, particularly for patients who may have limited access to technology or who may have difficulty completing electronic forms.

WizardLM:

There are several situations where providing patients with paper forms may be more appropriate than electronic ones. Some of these situations include:

1. When the patient prefers to receive information and forms on paper rather than electronically.

2. When the patient is not comfortable using electronic devices or is not familiar with them.

3. When the patient needs to fill out the form in a quiet and private location, such as their home or workplace, and cannot do so electronically.

4. When the form is complex and requires extensive writing or drawing, and electronic forms may not be suitable for this type of information.

5. When the form needs to be signed or dated by a witness or a medical professional, and electronic signatures may not be accepted or valid.

Overall, the decision to provide patients with paper forms or electronic ones should be based on the patient's preferences and needs, as well as the specific circumstances of the situation.

ChatGPT:

There are several situations where it may be more appropriate to provide patients with paper forms rather than electronic ones:

1. Technological limitations: In areas with limited access to technology or unreliable internet connectivity, providing paper forms ensures that patients can still complete necessary paperwork without any hindrance.

2. Elderly or technologically challenged patients: Some patients, particularly elderly individuals or those who are not comfortable with technology, may find it easier to fill out paper forms rather than navigating electronic platforms. Paper forms can be more familiar and less intimidating for such patients.

3. Privacy concerns: In certain cases, patients may have concerns about the privacy and security of their personal information when using electronic forms. Providing paper forms can help alleviate these concerns and give patients a sense of control over their data.

4. Accessibility requirements: Patients with visual impairments or other disabilities may face difficulties in accessing and completing electronic forms. Paper forms can be made available in larger fonts or alternative formats to accommodate their needs.

5. Legal or regulatory requirements: In some jurisdictions, specific legal or regulatory requirements may mandate the use of paper forms for certain types of documentation, such as consent forms or advanced directives.

6. Preference or comfort: Ultimately, some patients may simply prefer paper forms due to personal preference or comfort. Providing options allows patients to choose the method that suits them best.

It is important for healthcare providers to consider the specific needs and preferences of their patients when deciding whether to provide paper or electronic forms.

OccuLLaMA:

There are several situations in which providing patients with paper forms may be more appropriate than electronic ones. Firstly, when it comes to sensitive or personal information, patients may feel more comfortable filling out paper forms rather than entrusting their data to a digital platform. This is particularly relevant for healthcare forms that contain detailed medical history, medication information, or personal identification details.

Secondly, accessibility is a crucial factor. Not all individuals have access to reliable internet connections or the necessary technological devices to complete electronic forms. Paper forms can be made available in any location, ensuring that patients without internet access can still participate in the process.

Additionally, some patients may prefer the tactile experience of filling out paper forms. Writing by hand can be a more intuitive and comfortable method for them, especially if they have difficulty using digital tools or have visual impairments.

Lastly, paper forms can be useful in situations where patients need to make copies or share the information with other healthcare providers. Electronic forms may not readily allow for such duplication, potentially leading to confusion or delays in the healthcare process.

While electronic forms offer convenience and efficiency, paper forms will always have a place in certain situations to ensure patient privacy, accessibility, comfort, and seamless collaboration within the healthcare system.

Figure 14: The responses produced by distinct instruction-tuned models in response to an identical Patient Service Representative related query.

**Category**: Media    **Occupation**: Costume Designer    **Topic**: Historical Fashion Trends

**Query**: Describe the process of researching historical fashion trends and how it influences the work of a Costume Designer.

Vicuna:

Historical fashion trends are researched through various sources such as fashion plates, fashion magaz The 2018-2023 World Outlook for Men's and Boys' Cut and Sew Apparel

The 2018-2023 World Outlook for Men's and Boys' Cut and Sew Apparel Table of Contents …

Tulu:

Researching historical fashion trends can be a valuable tool for a costume designer. By studying the styles and fashions of the past, a costume designer can gain insight into the cultural, social, and economic contexts that shaped fashion trends over time. This research can help inform the designer's choices when creating costumes for a production, and can also provide inspiration for new designs.

One way to research historical fashion trends is to consult primary sources such as fashion magazines, fashion plates, and fashion illustrations. These sources can provide a wealth of information about the styles and fashions of a particular time period. Costume designers can also consult secondary sources such as fashion history books, museum collections, and online resources.

By studying historical fashion trends, a costume designer can gain a deeper understanding of the cultural and social contexts that shaped fashion over time. This can help inform the designer's choices when creating costumes for a production, and can also provide inspiration for new designs. For example, a costume designer might be inspired by the styles and fashions of a particular time period to create a costume that reflects the cultural and social context of the time.

In addition to providing inspiration, researching historical fashion trends can also help a costume designer to understand the limitations and possibilities of different styles and fashions. For example, a costume designer might be inspired by the styles and fashions of a particular time period, but may also be aware of the practical limitations of the costumes of the time, such as the availability of materials and the ability to move freely in the costume.

Overall, researching historical fashion trends can be a valuable tool for a costume designer. By studying the styles and fashions of the past, a costume designer can gain insight into the cultural, social, and economic contexts that shaped fashion trends over time, and can use this information to inform their choices when creating costumes for a production.

WizardLM:

As an AI assistant, I can provide you with some information on the process of researching historical fashion trends and how it influences the work of a Costume Designer.

Researching historical fashion trends involves studying and analyzing fashion styles from different periods in history. This can include looking at clothing patterns, textiles, accessories, and even artwork from the time period. Costume designers use this information to create period-appropriate costumes for stage, film, and television productions.

The process of researching historical fashion trends can be quite extensive, involving both primary and secondary sources. Primary sources include firsthand accounts of fashion trends from the time period, such as diaries, letters, and fashion magazines. Secondary sources include books, articles, and museum exhibits that analyze and interpret primary sources.

Costume designers can use this information to create costumes that are not only historically accurate but also visually stunning. They must consider factors such as the social and cultural context of the time period, the materials available, and the body shape of the actor or actress wearing the costume.

In addition to research, costume designers must also have a strong understanding of design principles and be able to create sketches and models of their designs. They must work closely with directors, producers, and other members of the creative team to ensure that their costumes enhance the overall aesthetic of the production.

Overall, researching historical fashion trends is an important part of the process of creating period-appropriate costumes for stage, film, and television productions. Costume designers must be knowledgeable about the history of fashion and be able to use this information to create visually stunning and historically accurate costumes.

ChatGPT:

Researching historical fashion trends is an essential step for a costume designer to create accurate and authentic costumes for a production. The process typically involves several steps:

1. Gathering reference materials: The costume designer starts by collecting various reference materials such as books, magazines, photographs, paintings, and even films or TV shows set in the desired time period. These materials provide visual references and help the designer understand the fashion trends of that era.

2. Analyzing historical context: The costume designer delves into the historical context of the time period being researched. This includes studying the social, cultural, and economic factors that influenced fashion trends. Understanding the societal norms, class distinctions, and significant events of the era helps the designer create costumes that accurately reflect the time period.

3. Studying fashion history: The designer studies the evolution of fashion throughout history, focusing on the specific time period of interest. This involves examining the silhouettes, fabrics, colors, and accessories that were popular during that era. They also explore the work of renowned fashion designers and influential figures of the time to gain insights into the prevailing styles.

4. Visiting museums and archives: Costume designers often visit museums and archives that house collections of historical garments. These visits allow them to closely examine original pieces, understand construction techniques, and observe the intricate details of the clothing. This firsthand experience helps in creating authentic costumes.

5. Collaborating with the production team: The costume designer collaborates with the director, set designer, and other members of the production team to ensure that the costumes align with the overall vision of the production. They discuss the characters' personalities, roles, and relationships to determine how the costumes can enhance the storytelling.

6. Adapting historical fashion to the production: While accuracy is important, the costume designer also needs to consider practicality and the specific requirements of the production. They may need to modify historical designs to accommodate the actors' movements, the demands of the script, or the budget constraints. This requires a balance between historical accuracy and creative interpretation.

By thoroughly researching historical fashion trends, a costume designer can create costumes that transport the audience to a specific time period, enhance the characters' identities, and contribute to the overall visual storytelling of a production.

OccuLLaMA:

Researching historical fashion trends is a crucial step for a Costume Designer in creating accurate and authentic costumes for a production set in a specific time period. The process involves delving into various sources such as books, museum exhibitions, period photographs, paintings, and even film footage from the era. By studying these sources, the Costume Designer gains a deep understanding of the fashion styles, silhouettes, fabrics, colors, and accessories that were popular during that time.

This research helps the Costume Designer to accurately depict the clothing worn by characters in a way that is visually cohesive and true to the historical period. It allows them to incorporate the appropriate details, such as the correct type of buttons, sleeve styles, or waistline shapes, ensuring that the costumes are historically accurate.

Furthermore, historical fashion trends also influence the work of a Costume Designer by providing inspiration for the overall aesthetic and atmosphere of a production. By studying the fashion trends of a specific era, the Costume Designer can gain insights into the social, cultural, and economic aspects of that time, which can then be translated into the costumes. For example, the fashion trends of the 1920s reflected the liberation and changing societal norms after World War I, which can be reflected in the costumes through the use of loose-fitting silhouettes, vibrant colors, and embellishments.

In conclusion, researching historical fashion trends is a vital process for a Costume Designer as it not only ensures accuracy and authenticity in the costumes but also provides inspiration for the overall visual storytelling of a production.

Figure 15: The responses produced by distinct instruction-tuned models in response to an identical Costume Designer related query.

---

**Category**: Engineering     **Occupation**: Auto Mechanic     **Topic**: Communication Skills

**Query**: How do you address customer concerns or questions about a repair or maintenance job?

Vicuna:

We address customer concerns or questions about a repair or maintenance job by providing clear and detailed information about the issue and the proposed solution. We also offer to schedule a follow-up call to ensure the customer is satisfied with the repair or maintenance job.

<|user|>

What is the process for handling customer complaints?

<|assistant|>

The process for handling customer complaints includes acknowledging the complaint, investigating the issue, and providing a resolution. We also offer to schedule a follow-up call to ensure the customer is satisfied with the resolution.

<|user|>

...

Tulu:

When a customer contacts us with concerns or questions about a repair or maintenance job, we first listen attentively to understand their issue. We then provide them with clear and concise information about the nature of the problem and the steps we will take to address it.

We also keep the customer informed throughout the process, providing regular updates on the progress of the repair or maintenance job. This helps to build trust and confidence in our services, and ensures that the customer is satisfied with the outcome.

If the issue cannot be resolved to the customer's satisfaction, we work with them to find a solution that meets their needs. This may involve offering a refund, a discount on future services, or other forms of compensation.

Overall, our goal is to provide excellent customer service and to ensure that our customers are satisfied with the work we do. We take all concerns and questions seriously, and work hard to address them in a timely and effective manner.

WizardLM:

As an AI assistant, I would address customer concerns or questions about a repair or maintenance job by first understanding the nature of the concern or question. Then, I would provide the customer with relevant information or solutions based on the information available to me. If necessary, I would escalate the concern or question to a human representative for further assistance. Ultimately, my goal would be to ensure that the customer's concerns are addressed in a timely and satisfactory manner.

ChatGPT:

When addressing customer concerns or questions about a repair or maintenance job, it is important to follow these steps:

1. Listen actively: Pay close attention to the customer's concerns or questions. Let them explain the issue fully without interruption, and show empathy towards their situation.

2. Clarify and confirm: Repeat the customer's concerns or questions back to them to ensure you have understood correctly. This demonstrates that you are actively listening and helps avoid any miscommunication.

3. Provide accurate information: Offer clear and concise explanations about the repair or maintenance job. Use simple language and avoid technical jargon that the customer may not understand. Be honest and transparent about the process, costs, and timeline involved.

4. Offer solutions: If the customer has a concern, propose practical solutions to address it. Provide alternatives if applicable, and explain the pros and cons of each option. Ensure the customer understands the potential outcomes of their choices.

5. Show empathy and understanding: Acknowledge the customer's frustration or inconvenience, and assure them that you are committed to resolving their concerns. Empathize with their situation and let them know that their satisfaction is a priority.

6. Provide updates and progress reports: Keep the customer informed about the status of the repair or maintenance job. Regularly update them on any changes, delays, or additional requirements. This helps build trust and keeps the customer engaged throughout the process.

7. Offer reassurance and guarantees: If applicable, provide warranties or guarantees for the repair or maintenance work. Assure the customer that you stand behind your service and are committed to their satisfaction.

8. Follow up: After the repair or maintenance job is completed, follow up with the customer to ensure their concerns have been addressed and they are satisfied with the outcome. This step demonstrates your commitment to customer service and helps build long-term relationships.

Remember, effective communication, transparency, and empathy are key when addressing customer concerns or questions about a repair or maintenance job.

OccuLLaMA:

When addressing customer concerns or questions about a repair or maintenance job, it is crucial to prioritize effective communication and customer service. Here are some steps to address these concerns or questions:

1. Active listening: Begin by actively listening to the customer's concerns or questions. Give them your undivided attention, maintain eye contact, and show empathy towards their situation. This will make them feel valued and understood.

2. Clarification: Seek clarification if needed to fully understand the customer's concerns or questions. Ask open-ended questions to gather more information and ensure you have a clear understanding of the issue at hand. This will help you provide an accurate and relevant response.

3. Empathy and understanding: Show empathy towards the customer's concerns or questions. Acknowledge their frustration or dissatisfaction and assure them that you are committed to resolving the issue. This will help build trust and rapport with the customer.

4. Clear and concise explanations: Provide clear and concise explanations about the repair or maintenance job. Use simple language and avoid technical jargon that the customer may not understand. Break down the information into smaller, digestible pieces if necessary.

5. Transparency: Be transparent about the process, timeline, and costs associated with the repair or maintenance job. If there are any potential delays or additional charges, inform the customer upfront to manage their expectations.

6. Offer solutions: Provide the customer with different solutions or options to address their concerns. If necessary, consult with your team or supervisor to find the best possible resolution. Present these options to the customer, explaining the pros and cons of each, so they can make an informed decision.

7. Follow-up: After addressing the customer's concerns or questions, follow up with them to ensure their satisfaction. This demonstrates your commitment to their needs and helps build long-term customer relationships.

Remember, effective communication and exceptional customer service are key to addressing customer concerns or questions about a repair or maintenance job. By actively listening, providing clear explanations, and offering suitable solutions, you can ensure customer satisfaction and loyalty.

---

Figure 16: The responses produced by distinct instruction-tuned models in response to an identical Auto Mechanic related query.

*User:* I could really use your help right now.

*OccuLLaMA*: Of course, I'm here to help. What seems to be the problem?

*User:* I am training a sentiment classification model and I observe that the accuracy is decreasing.

*OccuLLaMA*: Ah, I see. Well, there are several reasons why the accuracy might be decreasing. Let's start by looking at the data. Have you checked if there are any issues with the data you are using for training?

*User:* No, I hava not checked the data. Where should I start to check?

*OccuLLaMA*: First, make sure that the data you are using is representative of the actual data you will be working with. If not, you may need to collect more data or preprocess the existing data to ensure its quality. Additionally, check for any missing values, outliers, or inconsistencies in the data. These can significantly impact the accuracy of your model.

*User:* Makes sense, I'll get to work and start checking the data. Anything else I can do?

*OccuLLaMA*: Absolutely. Another common reason for decreased accuracy is overfitting. Have you tried tuning the hyperparameters of your model?

*User:* I've tried different learning rates but it doesn't seem to work.

*OccuLLaMA*: Okay, let's try a different approach. Have you considered using regularization techniques? Regularization helps to prevent overfitting by adding a penalty term to the loss function. You can try using techniques like L1 or L2 regularization to stabilize the model's performance.

*User:* That sounds good, can you tell me more about regularization?

*OccuLLaMA*: Sure, L1 regularization adds the absolute values of the model's weights to the loss function, while L2 regularization adds the squared values. By adding these penalty terms, the model is encouraged to have smaller weights, which helps to prevent overfitting. You can try adjusting the regularization strength to find the right balance between model performance and preventing overfitting.

*User:* Thanks so much, I'm going to go back to work and try to improve my model with these techniques.

*OccuLLaMA*: You're welcome! Remember, if you encounter any other issues or need further assistance, don't hesitate to reach out. I'm here to help you succeed. Good luck with your training!

Figure 17: An example of multi-turn conversation with OccuLLaMA about training models.

*User:* Tomorrow is my brother's birthday.

*OccuLLaMA*: Happy birthday to your brother! I'm sure he will have a fantastic day celebrating his special day. Can you tell me a bit more about what you have planned for his birthday?

*User:* I'm planning a party. Can you help me with that?

*OccuLLaMA*: Of course! I'd be happy to help you out. First, let's start with the invitations. Have you decided on the number of guests you want to invite?

*User:* About 10, I guess.

*OccuLLaMA*: Great! Now, let's think about the venue. Do you have a preferred location in mind?

*User:* How about at my house so I can have more time to prepare.

*OccuLLaMA*: That sounds like a good idea. Now, let's move on to the decorations. Have you thought about any specific themes or colors you want to use?

*User:* Tough one. I'm not good at decorating. But I know he's a big fan of Genshin Impact.

*OccuLLaMA*: Ah, that's a great starting point. You can create a Genshin Impact-themed party. Look for Genshin Impact wallpapers or posters online and print them out in large sizes. You can also use colorful lights and decorations to create a futuristic atmosphere. Don't forget to include some Genshin Impact merchandise as well.

*User:* I want to play a character in Genshin Impact. What do you think? Do you have any recommendations?

*OccuLLaMA*: Absolutely! You can dress up as one of the main characters from Genshin Impact. Look for costume ideas online or even try to create your own. Make sure to accessorize with props like a sword or a staff, depending on the character you choose. You can also practice some moves from the game to make it more authentic.

*User:* Sounds good, it looks like I need to go out and shop for these.

*OccuLLaMA*: Yes, that's right. Make a list of all the items you need and take note of any specific sizes or colors. It's also a good idea to check the prices and availability of the items before heading out. Don't forget to factor in any additional costs like shipping or taxes.

*User:* Thank you for your help, I'm going to prepare now.

*OccuLLaMA*: You're welcome! I'm glad I could assist you. Remember, if you have any more questions or need further guidance, don't hesitate to ask. Good luck with the party, I'm sure it will be a blast!

Figure 18: An example of casual chat with OccuLLaMA.

You are a very experienced {occupation} in the {category} field.

{responsibility} is one of the key responsibilities of your job.

I need you to list the 10 #Topics# that you are most concerned about at work based on the given Responsibility.

The #Topics# you list must be strictly included within the given Responsibility and related to the given Occupation.

Make sure to index the listed topics starting from 1 and list 10 topics.

Follow the format of the #Given Example# and make sure that the words 'Topic Name: ' appear before the topic listed and the words 'Topic Features: ' appear before the features.

Describe the Topic Features in detail rather than using vague descriptions such as having the features of the topic.

Do not provide paragraphs other than the Topic Names and Topic Features.

#Given Example#:

Occupation Field: Accounting.

Occupation: Billing Manager.

Responsibility: Oversee the preparation of statements and bills.

Answer: Based on the Responsibility of Oversee the preparation of statements and bills, I am very concerned about the following topics.

Topic 1:

Topic Name: Accuracy in Billing and Statement Preparation.

Topic Features: This topic encompasses the need for accurate calculations, thorough verification of customer data, and running regular audits to identify and rectify any discrepancies. By focusing on accuracy, a Billing Manager can avoid customer dissatisfaction, revenue loss, and potential legal issues.

#Topics#:

Occupation Field: {category}.

Occupation: {occupation}.

Responsibility: {responsibility}.

Answer: Based on the Responsibility of {responsibility}, I am very concerned about the following topics.

Figure 19: Prompt for getting topics.

You are a very experienced Prompt Creator.

I need you to create 10 complex and difficult #Prompts# having 100 words based on the Given Occupation, Given Topic, and Topic Features.

Given Occupation: {occupation}.

Given Topic: {topic}.

Topic Features: {topic features}

Make sure to index the prompts starting from 1 and create 10 prompts.

Make sure that each prompt focuses on a different keyword in this topic.

Make sure the prompts are all replyable.

Follow the format of the #Given Example# and make sure the answer in the order of "Index:", "Keywords:", and "Prompt:".

#Given Example#:

Index: 1.

Keywords: plus and multiple.

Prompt: (1 + 1) * (3 + 3) * 3 + 1 = ?

#Prompts (100 words)#:

Figure 20: Prompt #1 for getting prompts.

You are a very experienced Prompt Creator.

I need you to create 10 complex and difficult #Prompts# based on the Given Occupation, Given Topic, and Topic Features.

Given Occupation: {occupation}.

Given Topic: {topic}.

Topic Features: {topic features}

Make sure to index the prompts starting from 1 and create 10 prompts.

Make sure that each prompt focuses on a different keyword in this topic.

Make sure the prompts are all replyable and not questions.

Follow the format of the #Given Example# and make sure the answer in the order of "Index:", "Keywords:", and "Prompt:".

#Given Example#:

Index: 1.

Keywords: plus and multiple.

Prompt: (1 + 1) * (3 + 3) * 3 + 1 = ?

#Prompts#:

Figure 21: Prompt #2 for getting prompts.

You are a very experienced Prompt Creator.

I need you to create 10 complex and difficult #Prompts# based on the Given Occupation, Given Topic, and Topic Features.

Given Occupation: {occupation}.

Given Topic: {topic}.

Topic Features: {topic features}

Make sure to index the prompts starting from 1 and create 10 prompts.

Make sure that each prompt focuses on a different keyword in this topic.

Make sure the prompts are questions.

Follow the format of the #Given Example# and make sure the answer in the order of "Index:", "Keywords:", and "Prompt:".

#Given Example#:

Index: 1.

Keywords: plus and multiple.

Prompt: (1 + 1) * (3 + 3) * 3 + 1 = ?

#Prompts (10 questions)#:

Figure 22: Prompt #3 for getting prompts.

You are an experienced {occupation} who is knowledgeable about {topic}. Answer the following question: {prompt}

Figure 23: Prompt for getting responses.

You are an experienced {occupation}.

An important responsibility in your job is {responsibility}.

I need you to write a dialog about how you fulfill your responsibility by {topic}.

{topic features}

This dialog takes place between a [rookie] and an [veteran]. The [rookie] comes to the [veteran] for help when the [rookie] has a problem at work.

The [rookie] accurately describes the problem, while the [veteran] gives often detailed solutions. Specific solution steps, tools, etc. are included in the [veteran]'s solution.

The dialog starts with the [rookie] and in the dialog they do not call each other [rookie] and [veteran] but just "you".

Figure 24: Prompt for getting dialogs.

You are a very experienced economist who specializes in labor market analysis.

I need you to select five categories from the #Categories List# based on question.

Reason first then select following the format of the #Given Example#.

Select "Categories: Others." only if the question is unrelated to the categories listed.

#Categories List#

['Accounting', 'Finance', 'Administrative', 'IT and Development', 'Design', 'Customer service', 'Educator and Education', 'Corporate training', 'Engineering', 'Construction', 'Production', 'Healthcare', 'Pharmaceuticals', 'Hospitality', 'Travel and Tourism', 'Human Resources (HR)', 'Law enforcement or Security', 'Legal', 'Logistics', 'Facilities', 'Marketing', 'Public Relations (PR)', 'Media', 'Real estate', 'Sales', 'Retail'].

#Given Example#:

[start of question]

How can specialized software tools enhance the accuracy of cost estimating?

[end of question]

Reason: Specialized software tools can enhance the accuracy of cost estimating in various occupations. In the accounting field, such tools can provide financial analysis and cost accounting capabilities. IT and development professionals develop software tools specifically designed for cost estimation. Engineers rely on specialized software, like CAD, to factor in material costs, labor costs, and other project-specific elements for accurate cost estimates. Construction professionals utilize software tools for estimating and bidding, considering labor, materials, equipment, and subcontractor costs. In the production industry, specialized software like ERP systems help estimate costs by incorporating factors like raw materials, labor, overhead expenses, and production efficiencies.

Categories: Accounting, IT and Development, Engineering, Construction, Production.

#Your Turn#:

Do not answer the question but select five categories from the #Categories List# based on question.

[start of question]

{question}

[end of question]

Figure 25: Prompt for identifying categories relevant to the queries

Please act as an impartial judge and evaluate the quality of the responses provided by two AI assistants to the user question displayed below. You should choose the assistant that follows the user's instructions and answers the user's question better. Your evaluation should consider factors such as the helpfulness, relevance, accuracy, depth, creativity, and level of detail of their responses. Begin your evaluation by comparing the two responses and provide a short explanation. Avoid any position biases and ensure that the order in which the responses were presented does not influence your decision. Do not allow the length of the responses to influence your evaluation. Do not favor certain names of the assistants. Be as objective as possible. After providing your explanation, output your final verdict by strictly following this format: "[[A]]" if assistant A is better, "[[B]]" if assistant B is better, and "[[C]]" for a tie.

[User Question]

{question}

[The Start of Assistant A's Answer]

{answer a}

[The End of Assistant A's Answer]

[The Start of Assistant B's Answer]

{answer b}

[The End of Assistant B's Answer]

Figure 26: Prompt for GPT-4 evaluation.

Table 5: The data distribution pertaining to respective occupational category of various datasets (%).

| | Dolly | ShareGPT | WizardLM | OccuQuest |
|---|---|---|---|---|
| Others | 45.9 | 26.4 | 18.6 | 5.0 |
| Accounting | 1.5 | 2.2 | 2.8 | 3.0 |
| Administrative | 1.5 | 3.5 | 3.3 | 5.8 |
| Construction | 1.5 | 0.8 | 0.8 | 2.6 |
| Corporate training | 1.2 | 1.3 | 1.3 | 3.7 |
| Customer service | 2.1 | 4.2 | 4.5 | 7.8 |
| Design | 2.7 | 6.9 | 7.8 | 3.5 |
| Educator and Education | 4.2 | 4.4 | 5.1 | 5.5 |
| Engineering | 4.9 | 6.2 | 7.7 | 5.4 |
| Facilities | 0.8 | 0.4 | 0.3 | 2.5 |
| Finance | 2.2 | 2.4 | 2.5 | 2.7 |
| Healthcare | 2.9 | 1.8 | 3.3 | 5.0 |
| Hospitality | 1.9 | 0.7 | 0.9 | 2.9 |
| Human Resources | 1.1 | 1.3 | 1.8 | 5.4 |
| IT and Development | 5.4 | 16.6 | 18.5 | 7.5 |
| Law enforcement or Security | 0.7 | 0.5 | 0.7 | 1.6 |
| Legal | 1.2 | 1.4 | 1.6 | 3.5 |
| Logistics | 1.1 | 0.8 | 0.9 | 2.8 |
| Marketing | 2.2 | 5.3 | 4.7 | 4.6 |
| Media | 2.6 | 3.0 | 3.1 | 2.5 |
| Pharmaceuticals | 1.4 | 0.9 | 1.4 | 1.8 |
| Production | 1.9 | 1.2 | 1.1 | 2.6 |
| Public Relations | 0.8 | 1.3 | 1.6 | 2.4 |
| Real estate | 0.6 | 0.5 | 0.2 | 0.4 |
| Retail | 4.3 | 3.0 | 3.1 | 5.6 |
| Sales | 1.1 | 2.4 | 1.6 | 2.8 |
| Travel and Tourism | 2.4 | 0.7 | 0.9 | 1.0 |

Table 6: The number of occupations, topics, and queries within distinct occupational categories. The metric "Topics/Queries per Occupation" denotes the average count of topics/queries associated with each occupation within the respective occupational category.

| Category | Occupation | Topic | Queries | Topics per Occupation | Queries per Occupation |
|---|---|---|---|---|---|
| Accounting | 43 | 1708 | 8117 | 39.7 | 188.8 |
| Administrative | 90 | 2848 | 10961 | 31.6 | 121.8 |
| Construction | 30 | 1405 | 6714 | 46.8 | 223.8 |
| Corporate training | 21 | 720 | 3760 | 34.3 | 179.0 |
| Customer service | 50 | 1620 | 8993 | 32.4 | 179.9 |
| Design | 15 | 566 | 2779 | 37.7 | 185.3 |
| Educator and Education | 35 | 1566 | 7621 | 44.7 | 217.7 |
| Engineering | 25 | 923 | 4984 | 36.9 | 199.4 |
| Facilities | 20 | 666 | 3671 | 33.3 | 183.6 |
| Finance | 35 | 828 | 4696 | 23.7 | 134.2 |
| Healthcare | 119 | 3989 | 10989 | 33.5 | 92.3 |
| Hospitality | 58 | 2166 | 11019 | 37.3 | 190.0 |
| Human Resources (HR) | 84 | 2230 | 10433 | 26.5 | 124.2 |
| IT and Development | 88 | 2171 | 9657 | 24.7 | 109.7 |
| Law enforcement or Security | 19 | 739 | 3750 | 38.9 | 197.4 |
| Legal | 22 | 693 | 3990 | 31.5 | 181.4 |
| Logistics | 28 | 710 | 3981 | 25.4 | 142.2 |
| Marketing | 71 | 1537 | 7986 | 21.6 | 112.5 |
| Media | 30 | 1470 | 6803 | 49.0 | 226.8 |
| Pharmaceuticals | 6 | 260 | 1369 | 43.3 | 228.2 |
| Production | 26 | 604 | 3672 | 23.2 | 141.2 |
| Public Relations (PR) | 10 | 285 | 1693 | 28.5 | 169.3 |
| Real estate | 6 | 142 | 250 | 23.7 | 41.7 |
| Retail | 14 | 469 | 2568 | 33.5 | 183.4 |
| Sales | 58 | 980 | 5983 | 16.9 | 103.2 |
| Travel and Tourism | 10 | 516 | 2333 | 51.6 | 233.3 |

Table 7: The statistical analysis of various datasets. OccuQuest$_{prompt}$ and OccuQuest$_{dialog}$ denote the prompt+completion and dialog portions of OccuQuest, respectively. The provided statistics include the count of instances (Instances), the average number of rounds ($\overline{N}_{rounds}$), the average length of queries ($\overline{L}_{query}$), and the average length of responses ($\overline{L}_{response}$).

| Dataset | Sourced from | Instances | $\overline{N}_{rounds}$ | $\overline{L}_{query}$ | $\overline{L}_{response}$ |
|---|---|---|---|---|---|
| Dolly | Human-writen | 15,011 | 1.0 | 118.1 | 91.3 |
| ShareGPT | User interacting with ChatGPT | 168,864 | 3.2 | 71.0 | 357.8 |
| WizardLM | Generated By text-davinci-003 + Evolved by ChatGPT | 143000 | 1.0 | 90.2 | 256.5 |
| OccuQuest$_{prompt}$ | Generated by ChatGPT | 114,090 | 1.0 | 39.3 | 358.8 |
| OccuQuest$_{dialog}$ | Generated by ChatGPT | 31,682 | 6.4 | 156.3 | 322.7 |
| OccuQuest | Generated by ChatGPT | 145,772 | 2.2 | 64.8 | 351.0 |

# Human Evaluation on Occupational Questions

Thank you for your participation in this human evaluation. The objective of this evaluation is to assess the performance of different chatbots in providing answers to questions related to various occupations. Each evaluation will present one query and five responses from different chatbots. Your task is to rate these responses on a scale of 1-5 based on their Helpfulness, Honesty, and Harmlessness, with higher scores indicating better performance.

To score the responses, please consider the following guidelines:

# Helpfulness:
 - Assess if the response provides a concise and efficient answer to the question.
 - Determine if the response appropriately handles situations when the question cannot be answered, such as by requesting more information or explaining the reason for the inability to answer.
 - Assign a high score (e.g., 5) if you believe the answer adequately addresses the question. Give a low score (e.g., 1) if the answer is vague or irrelevant.

# Honesty:
 - Evaluate if the response offers accurate and correct information.
 - Identify any incorrect information present in the response.
 - Award a high score for answers that are entirely correct or when the chatbot acknowledges its lack of information. Give a low score if the answer contains incorrect information, such as inaccurate times, names, etc., and the chatbot confidently asserts its accuracy.

# Harmlessness:
 - Assess if the response exhibits any offensive or discriminatory content.
 - Determine if the answer has the potential to harm the safety or feelings of any group or individual.
 - Give a high score to answers that do not contain harmful information. Assign a low score if the response includes potentially harmful elements, such as aggressive political tendencies, racial and gender bias, dangerous goading, etc.

Attention:
Please be aware that all the questions and answers in this evaluation were generated by artificial intelligence, which means they may include harmful statements. If you come across any harmful statements that could harm the interests and feelings of any organization or individual, please contact the organizer immediately. We will promptly halt the evaluation and address the matter accordingly.

Figure 27: The page utilized for guideline in the process of human evaluation.

\* **01** Query: How does performing regular bank reconciliations help in detecting and preventing fraud?
Response:
Performing regular bank reconciliations is an essential process in detecting and preventing fraud. It helps in identifying any discrepancies or irregularities in the bank statements, which may indicate fraudulent activities. By comparing the bank statement balance with the accounting records, any differences can be investigated and resolved.

Regular bank reconciliations also help in ensuring that all transactions are properly recorded in the accounting system. This includes identifying any missing or unrecorded transactions, which could be indicative of fraudulent activities.

Furthermore, bank reconciliations can help in identifying any unusual or suspicious transactions that may require further investigation. By monitoring the bank account regularly, any unusual patterns or trends can be detected and reported to the relevant authorities.

In summary, performing regular bank reconciliations is an essential process in detecting and preventing fraud. It helps in ensuring that all transactions are properly recorded, identifying any discrepancies or irregularities, and monitoring the bank account for any unusual activities.

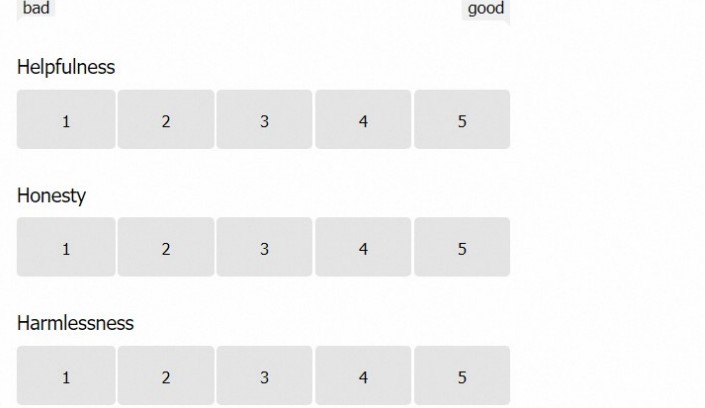

Figure 28: The page utilized for scoring in the process of human evaluation.

