# OpenReview forum: "OccuQuest: Mitigating Occupational Bias for Inclusive Large Language Models"
_ICLR.cc/2024/Conference — ICLR 2024 Conference Withdrawn Submission_

### Official Review · Reviewer_fj7f · 2023-10-29

**Soundness:** 2 fair
**Presentation:** 3 good
**Contribution:** 3 good
**Rating:** 5
**Confidence:** 4

**Summary:**

This paper propose an instruction-tuning dataset OCCUQUEST focusing on mitigating the issue of occupational bias in LLMs and essentially enhancing the instruction-tuned LLMs to generate helpful responses to professional queries from practitioners in specific fields. This paper fine-tunes the LLaMA-7B model on OccuQuest and shows competitive results in comparison with other baselines(Vicuna, Tulu, WizardLM and ChatGPT). This paper also proposes ProLLaMA, a series of LLaMA models that excel in answering questions from different occupations and perform well on the comprehensive abilities assessments.

**Strengths:**

1. This is the first dataset available that focuses on mitigating the issue of occupational bias in LLMs.
2. The submission is clear and detailed. The author describes the data set generation scheme completely, and fully demonstrates experiment settings which make this paper reproducible.
3. The experiment results are relatively competitive which means the proposed method has the potential to mitigate the occupational bias in LLMs.

**Weaknesses:**

1. This paper lacks objective evaluation in the performance of proposed dataset. When testing the OccuLLaMA, the three testing datasets are specially constructed by the author which is unfair for the compared baselines. As for the ProLLaMA, there is a mixture of OccuQuest and Tulu datasets in training phrase.
2. There might be hallucinations in generated dataset which is not considered in this paper.
3. The paper needs further analysis on whether there is mutual influence between occupations when the number of occupational categories increases significantly (over 1000 categories mentioned in the article) and after the samples are balanced.
4. Mixing OccuQuest with other datasets may improve the model's performance on MMLU, GSM8K, HumanEval, etc. Further experiments are needed to analyze the correlation between them. It's also necessary to check whether there is a possibility of evaluation data leakage when collecting data through ChatGPT.

**Questions:**

See points 3 and 4 in Weaknesses.

---

### Official Review · Reviewer_Y9X6 · 2023-11-01

**Soundness:** 2 fair
**Presentation:** 2 fair
**Contribution:** 2 fair
**Rating:** 3
**Confidence:** 3

**Summary:**

This paper introduces OccuQuest, which generates instruction-output data from ChatGPT prompted by a set of prompts consisting of occupations and responsibilities. They generated about 110k prompt completions and 39k dialogue data across 26 occupation categories. Experimental results show that the model trained on the OccuQuest dataset can outperform other 7B scale models on test sets (different split of OccuQuest and Quora) by human and GPT-4- based ELO evaluation. They also show that combining their training data with existing training data e.g., Tulu can lead to strong task performance.

Overall, I don't think this paper provides sufficient novelties--generating instruction-output training data has been actively studied and synthetically creating occupation-guided prompt queries only does not provide enough technical contributions to ICLR--and discrepancies between original motivations (to train an LM to generate helpful responses to professional queries from practitioners in specific fields) and proposed method / final evaluation protocols. See the detailed discussion in the Weakness section.

**Strengths:**

- Proposed method introduces a new instruction-tuning data creation method using a strong LM
- Experimental results on their test set shows that their method outperforms other 7b LMs.

**Weaknesses:**

### Soundness of prposed method and evaluations to enhance LM abilities in expert domains

The core motivation of this work is to enhance LMs' abilities of generating more helpful responses to professional domain queries.
>  However, existing instruction-tuning datasets suffer from occupational bias: the majority of data relates to only a few occupations, which hampers the instruction-tuned LLMs to generate helpful responses to professional queries from practitioners in specific fields. To mitigate this issue and promote occupation-inclusive LLMs, we create an instruction-tuning dataset named OccuQuest

I don't think the proposed method properly address this issue (unless careful human evaluation indicates that generated data is indeed high-quality) and the evaluation protocols are appropriate to assess the quality of model generations in those areas.

**Proposed method**
I don't agree that this approach, generating more training data from ChatGPT, can result in high-quality data to train reliable LMs in professional domains. Growing number of papers show that ChatGPT or GPT 0002/003 frequently hallucinate or fabricated responses to such professional queries or long-tail areas. Generating training data from ChatGPT for those professional queries can easily result in many factually incorrect responses, and smaller LMs trained on this distilled data can follow this trend. If the main focus of this work is to improve LM generation quality in such domains, I think the essential step is to conduct thorough expert analysis on generated queries, and check whether the queries are high-quality.

**Evaluation protocols**
The main evaluations are soley based on GPT-4 or human preference-based evaluations on the OccuQuest test sets created by the same protocol as the training dataset, or Quora queries. It is not surprsing that the model trained on OccuQuest train sets can obtain stronger performance on the in-domain test set, and I don't think getting higher win-rate on those datasets indicate whether the model indeeds performs better.

Moreover, the overall preference-based method may not be suitable for such usecases as (1) overall preference-based evaluations are often heavily influenced by superficial factors, and (2) human evaluators (or GPT-4) may not be fully reliable to assess the quality of generations in professional / expert domains.

(1) Recent work including the Tulu paper shows that GPT-4-based evaluations are often biased toward the longer response, and may not reflect the true quality of generation. To assess the quality of responses, especially in professional domains as authors claimed in the paper, more careful and fine-grained evaluations should be done.

(2) The human evaluations are also conducted by three annotators on 100 queries. Are the three annotators familiar with all of the domains of 26 occupations? If the annotators are not fully familiar with all target areas, the preference evaluation may not be reliable.
For instance, a recent ExpertQA paper (Malaviya et al., 2023) conducted fine-grained human evaluations from multiple aspects (e.g., evidence, responses, attribution, reliability of evidence) by hiring experts. [Xu et al. (2023)](https://arxiv.org/abs/2305.18201) also show that to evaluate the quality of long-form QA, human annotators need to have expertise in the area.

**LImited technical ontributions or novelties**

This work shows interesting empirical findings and exploration, but I don't think formulating occupation-centric queries and generating more instruction-output data from a strong LM are sufficient contributions for ICLR.

**Questions:**

- Did you conduct human analyis on created instruction-output training data by hiring domain experts?
- Did you conduct fine-grained human evaluations or GPT-4 evaluations to evaluate the model responses for the test sets?

This is not a question, but there are many of overlapping text in the Figures. Probably trying different configurations to avoid such overlap in Figure may help. Also Figure 6 occupation names are cut off and for some bars, I am not sure which occupation corresponds to the substring.

---

### Official Review · Reviewer_gJH9 · 2023-11-13

**Soundness:** 3 good
**Presentation:** 3 good
**Contribution:** 3 good
**Rating:** 5
**Confidence:** 4

**Summary:**

This paper contributes OccuQuest dataset that contains queries and responses about 1,013 occupations under 26 occupational categories collected from Workable and generated by prompting ChatGPT. Each item in the dataset contains the occupation name and occupation category (collected from Workable) and topic, topic features, queries, and responses (generated by prompting ChatGPT). The authors show that compared to existing instruction-tuning datasets (Dolly, ShareGPT, WizardLM), OccuQuest has a more balanced distribution of occupations. They fine-tune LLaMA on OccuQuest to obtain OccuLLaMA (which has higher performance compared to other LLaMA variants in answering occupational-related queries) and fine-tune LLaMA on OccuQuest and Tulu datasets to obtain ProLLaMA (which has good performance in answering both occupational-related queries and other comprehensive abilities).

**Strengths:**

The strength of the paper lies in its contribution of a new dataset containing occupational queries and responses that has more balanced distribution of occupations.

**Weaknesses:**

The major weaknesses of the paper are:
- The dataset is generated by a large language model (ChatGPT) and thus are prone to inaccuracies and hallucinations; and potentially other biases. Thus, even though the dataset is more balanced in terms of occupations, it may contain biases about occupations that can be propagated to other LLMs trained on the dataset.
- Unfortunately, the human evaluation of the dataset does not evaluate for these potential inaccuracies or biases in details, only asking human evaluators to measure helpfulness, honesty, and harmlessness on a scale from 1-5 and on a small sample of the dataset: only 100 samples of more than 100K instances in the dataset were evaluated.
- The majority of the evaluation itself was conducted using GPT-4, which may cause bias, since the dataset itself is generated by ChatGPT.

**Questions:**

What is the version of ChatGPT used to generate the dataset?